# Can goal-setting for patients with multimorbidity improve outcomes in primary care? Cluster randomised feasibility trial

John A Ford,[1] Elizabeth Lenaghan,[1] Charlotte Salter,[1] David Turner,[1] Alice Shiner,[1] Allan B Clark,[1] Jamie Murdoch,[1] Carole Green,[2] Sarah James,[1] Imogen Koopmans,[1] Alistair Lipp,[3] Annie Moseley,[2] Tom Wade,[1] Sandra Winterburn,[1] Nicholas Steel[1]

¹Faculty of Medicine and Health Sciences, University of East Anglia, Norwich, UK
²Patient and public involvement representative, Norwich, UK
³NHS England Midlands & East (East), Fulbourn, UK

**Correspondence to**
Dr Nicholas Steel;
n.steel@uea.ac.uk

## ABSTRACT

**Introduction** Goal-setting is recommended for patients with multimorbidity, but there is little evidence to support its use in general practice.

**Objective** To assess the feasibility of goal-setting for patients with multimorbidity, before undertaking a definitive trial.

**Design and setting** Cluster-randomised controlled feasibility trial of goal-setting compared with control in six general practices.

**Participants** Adults with two or more long term health conditions and at risk of unplanned hospital admission.

**Interventions** General practitioners (GPs) underwent training and patients were asked to consider goals before an initial goal-setting consultation and a follow-up consultation 6 months later. The control group received usual care planning.

**Outcome measures** Health-related quality of life (EQ-5D-5L), capability (ICEpop CAPability measure for Older people), Patient Assessment of Chronic Illness Care and healthcare use. All consultations were video-recorded or audio-recorded, and focus groups were held with participating GPs and patients.

**Results** Fifty-two participants were recruited with a response rate of 12%. Full follow-up data were available for 41. In the goal-setting group, mean age was 80.4 years, 54% were female and the median number of prescribed medications was 13, compared with 77.2 years, 39% female and 11.5 medications in the control group. The mean initial consultation time was 23.0 min in the goal-setting group and 19.2 in the control group. Overall 28% of patient participants had no cognitive impairment. Participants set between one and three goals on a wide range of subjects, such as chronic disease management, walking, maintaining social and leisure interests, and weight management. Patient participants found goal-setting acceptable and would have liked more frequent follow-up. GPs unanimously liked goal-setting and felt it delivered more patient-centred care, and they highlighted the importance of training.

**Conclusions** This goal-setting intervention was feasible to deliver in general practice. A larger, definitive study is needed to test its effectiveness.

### Strengths and limitations of this study

► General practitioners (GPs) and patients with multimorbidities both benefit from preparation before setting goals.

► Recruitment reached target levels in five of six practices, but the patient response rate of 12% means that a definitive study will need sufficient numbers of patients with multimorbidity.

► Existing measures of patient centred care are usually designed for a single specific treatment decision and were difficult to apply to goal-setting consultations, where several goals were discussed.

► The most relevant outcome measure for goal-setting was the Patient Assessment of Chronic Illness Care, which includes a subscale for goal-setting.

► Qualitative data from video-recorded consultations and focus groups were vital to understand how goal-setting was implemented in practice, and how acceptable it was to GPs and patients.

**Trial registration number** ISRCTN13248305; Post-results.

### INTRODUCTION

The rising number of long-term conditions and prescribed medications has increased the burden of treatment for patients.[1 2] People with multimorbidity (defined as two or more chronic conditions[2]) tend to have a lower quality of life and worse health than those with single conditions.[3] Medical outcomes that work well for relatively healthy patients (eg, blood pressure control or disease-free survival) may be inappropriate for patients with multimorbidity or severe disability,[4 5] and the use of current single-disease guidelines in this group can encourage harmful polypharmacy with resulting drug–drug and drug–disease interactions.[6]

The National Institute for Health and Care Excellence recommends an approach to care that takes account of multimorbidity by establishing patient goals, values and priorities.[7] Goal-setting is the sharing of realistic goals by health professionals and patients and agreement of the best course of action.[8] Goal-setting enables patients and doctors to focus healthcare on the outcomes that are most important to the patient. Examples of outcomes that matter to patients may include maintaining independence, undertaking paid or voluntary work, preventing adverse outcomes (eg, falls) and reducing treatment burden.[7] Despite the recommendation that health professionals should establish patient goals with individuals with multimorbidity, there is little evidence to support the use of goal-setting between general practitioners (GPs) and patients, and it is rarely used in primary care.[8–10] The goal-setting approach is more likely to be effective if it incorporates shared decision making, the process by which health professionals and patients make decisions together based on the best available evidence,[11] because the goals and actions agreed will be more patient-centred. The difference is that shared decision making is usually concerned with specific clinical treatment decisions, whereas goal-setting usually involves a wider discussion around ways to deliver outcomes that matter to the patient.

Goal-setting should be, but rarely is, an important element of the care planning process in the UK. For the purposes of this study, we define care planning as 'a conversation in which patients and clinicians agree on goals and actions for managing the patient's conditions'.[8] For patients with long term health conditions, personalised care planning has been found to improve physical and psychological health, in addition improving capability to self-manage, compared with usual care.[8] A recent systematic review highlighted the need for evidence exploring 'the effects of personalised care planning on goal-attainment, especially patient's personal goals as opposed to goals determined by clinicians or researchers'.[12]

Our goal-setting intervention was designed within the context of a national recommendation that the top 2% of patients at risk of unplanned hospital admission should have a care plan.[13] We wanted to find out if a consultation focused on goal-setting would improve outcomes for this patient group, compared with control consultations (the usual care planning process undertaken in UK primary care which rarely includes goal-setting). Before we could conduct a full trial to answer this question, we needed to answer questions about the feasibility of such a trial. We aimed to assess the feasibility of goal-setting for patients with multimorbidity, at high risk of hospital admission and eligible for a care planning consultation, with a view to undertaking a future definitive randomised controlled trial. Our objectives were to assess participant recruitment and retention, the acceptability of a goal-setting intervention to patients and GPs, the training needs of GPs, the content of control consultations, goal-setting and the feasibility of collecting relevant outcome measures.

## METHODS

We undertook a cluster randomised controlled feasibility trial of goal-setting compared with usual care in six general practices in the United Kingdom, with 6 months follow-up. Six months was long enough for patients and GPs to work towards the agreed goals, but not so long that the goals would have been forgotten. There were no significant changes to the protocol.[14] Participants were recruited between April and May 2017 and follow-up completed in February 2018.

General practices were invited via two emails through the East of England Clinical Research Network and recruited on a first-come first-served basis. To be eligible, practices had to be using risk stratification to identify patients at high risk of unplanned admission (eg, by participating in the Avoiding Unplanned Admissions Enhanced Service: proactive case finding and patient review for vulnerable people[13]), have at least one Good Clinical Practice trained GP and nurse, be able to nominate two GPs to attend the goal-setting training and not be a single handed practice. Practices were reimbursed for staff time and travel to undertake the research and deliver the intervention. Patients were eligible if they were aged 18 or over, identified as in the top 2% for risk of unplanned admission and diagnosed with at least two of 40 morbidities in Barnett's analysis of multimorbidity.[2] Patients were excluded if they were deemed to be unable to participate in goal-setting in the GP's professional opinion (eg, advanced dementia or acute psychosis), had received a care planning consultation in the previous 3 months, or required translation services to communicate verbally.

Practice administrators searched their electronic patient register according to the eligibility criteria, and a GP then checked the resulting patient list for exclusion criteria. Eligible patients were sent a letter of invitation and participant information leaflet, with the intention of recruiting 10 patients per practice. The number of eligible patients ranged from 47 to 124 and all were invited. The protocol allowed GPs to opportunistically invite patients they thought might be interested, however no patients were recruited through this process. A study researcher visited interested patients at home to discuss the study and obtain written informed consent.

The Norwich Clinical Trials Unit independently randomised three practices to goal-setting and three to control, by simple block randomisation using a 1:1 ratio and sealed opaque envelopes. Practices were randomised after at least 10 expressions of interest were received from patients. It was not possible to blind participants, health professionals or researchers due to the nature of the intervention, with the exception of the statistician undertaking the analysis, who was blinded to the allocation.

### Intervention

Both intervention and control practices identified two GPs to either attend the training and deliver goal-setting consultations or deliver control consultations, although in one intervention practice (Practice 3) only one GP

**Table 1** Baseline characteristics of participating practices and patients, by practice

| | Goal-setting | | | Control | | |
|---|---|---|---|---|---|---|
| | **Practice 1** | **Practice 2** | **Practice 3** | **Practice 4** | **Practice 5** | **Practice 6** |
| **Practice characteristics** | | | | | | |
| Practice rurality* | Village | Town and fringe | Town and fringe | Urban>10K | Urban>10K | Urban>10K |
| Patient population | 5000–9900 | 10 000–14 900 | 5000–9900 | >14 900 | 10 000–14 900 | 10 000–14 900 |
| IMD practice decile | 7 | 5 | 7 | 9 | 5 | 5 |
| Characteristics of participating GPs | n=2 Both male, partners and working part-time | n=2 One male, one female, both partners and working full-time | n=1 Male, partner working part-time | n=2 One male, one female, both partners, one working full-time and one part-time | n=2 both female, partners and working part-time | n=2 both female, partners and working part-time |
| Years qualified of participating GPs | GP014 >20 years; GP018=10– 20 years | GP025 <10 years; GP026=10– 20 years | GP038=10– 20 years | GP046 >20 years; GP047 >20 years | GP053 >20 years; GP055 >20 years | GP061=10– 20 years; GP067=10–20 years |
| **Practice recruitment** | | | | | | |
| Patients assessed for eligibility, n | 9067 | 14 845 | 6791 | 18 540 | 10 381 | 13 439 |
| Patients invited, n (% assessed) | 77 (0.8) | 108 (0.7) | 47 (0.7) | 108 (0.6) | 124 (1.2) | 86 (0.6) |
| Recruited, n (% invited)† | 11 (14.3) | 9 (8.3) | 4 (8.5) | 8 (7.4) | 10 (11.6) | 10 (11.6) |

Partner=GP with responsibility for the practice.
*ONS indicator 2011.[37]
†Does not include those on the reserve list (see figure 1).
GPs, general practitioners; IMD, Index of Multiple Deprivation (1=most deprived and 10=least deprived).

was able to attend. Therefore five participating GPs from practices allocated to goal-setting (see table 1) received training in a 3-hour experiential workshop, led by senior consultation skills tutors (CS and SW) and a GP with experience in communication skills training (AS). One other GP attended the training but withdrew prior to delivering the intervention for personal reasons. The training model we developed for goal-setting adapted relevant elements of the work of Elwyn and colleagues on shared decision making[15 16] and of patient-centred care in the leading training model in clinical communication (the Calgary Cambridge Guide[17]). Our model adopted a structured, patient-centred stepped approach. Steps included preparation, goal elicitation, assessing options, making goals smart, decision-making and evaluation. Following an introduction to the study, the training was mainly experiential to enable GPs to rehearse existing skills and integrate additional skills for facilitating the goal-setting process. Experiential methods included role-play, video analysis and interactive skill spotting. GPs were trained in groups of three and were given a detailed handbook in advance. The handbook contained information about the study and a 'how to' guide for goal-setting, including theoretical background and examples of goal-setting. The control group GPs received no training for this study and were asked to undertake a care planning consultation as they would usually do in routine clinical practice. This may have involved a national care planning template, which does not include goal-setting, from the Avoiding Unplanned Admissions Enhanced Service.[13]

A study researcher discussed goal-setting and the associated paperwork with participants during the face-to-face baseline visit, which lasted approximately 15 min. The researcher gave all patient participants a patient-held goal-setting sheet, with questions to consider prior to their consultation. The questions (online supplementary appendix 1) were:

▶ What are your goals? What is important to you? What do you really want to achieve over the next 6 months?
▶ Why are these goals important to you?
▶ What are the first steps you would like to take towards achieving this goal or goals?

The goal-setting consultations were held with the participating GPs even if they were different from the patient's usual GP. During the initial goal-setting consultation GPs, in partnership with participants, documented the goals which had been agreed. GPs then provided support, within their clinical expertise and with the help of other healthcare professionals, to help patients achieve their goals, for example by providing information on local groups and services. Participants in both the goal-setting and control groups had an initial consultation which lasted about 20 min, but only patients in the goal-setting arm were invited back for a follow-up consultation after 6 months to discuss their goal attainment.

### Data and statistical analysis

We collected quantitative and qualitative data to meet the feasibility study objectives. Data collected from patients during a researcher visit at baseline and 6 months were:

health-related quality of life (EQ-5D-5L[18]); capability (as measured through the five attributes of attachment, security, role, enjoyment and control in the ICEpop CAPability measure for Older people questionnaire (ICECAP-O)[19]) (ICEpop is the name of the UK MRC-funded programme through which the index was developed), cognition (General Practitioner assessment of Cognition scale (GPCOG)[20]) and patient centred care (Patient Assessment of Chronic Illness Care (PACIC)[21]). Data collected from the electronic patient record included age, sex and postcode Index of Multiple Deprivation (IMD) score (baseline only), medications on repeat prescription, diagnoses, achievement of relevant quality of care indicators in the Quality and Outcomes Framework[22] and primary and secondary care use (see the Health economic evaluation section for more details). Practice data were collected before randomisation and patient data were collected after.

GPs and patient participants were asked to complete an assessment of shared decision making during each consultation using the CollaboRATE scale[23] for patients and Dyadic OPTION scale[24] for GPs. GPs and patients in the goal-setting group were asked to discuss and complete a Goal Attainment Scaling (GAS-Light) questionnaire[25] (See online supplementary appendix 2) at the second consultation. Goal attainment was scored using the following system: −1=worse than expected, 0=no change, 1=partially attained, 2=as expected, 3=a little more and 4=a lot more than expected.

All initial consultations were video-recorded (n=41) or audio-recorded (n=4) and transcribed. Three team members scored the consultations using the observer OPTION measure to assess shared decision making.[26] One focus group was held with patients and one with GPs from the goal-setting group at the end of the 6 month follow-up period to discuss perspectives, experiences and overall acceptability of the goal-setting intervention. All patients in the intervention group were sent a letter of invitation to the focus group, except two who indicated at the researcher visit they did not want to take part. Both focus groups lasted about 90 min, were held at the university, guided by a topic guide, audio-recorded and transcribed. Patient or GP participants unable to attend the focus groups were interviewed by phone or face-to-face using the same topic guide.

We calculated the recruitment rate by practice and by randomisation group. Demographic variables were compared for those recruited and those not recruited. The characteristics of baseline consultations were summarised both by practice and by intervention group.

The change in outcome measures from baseline to follow-up was summarised using descriptive statistics by randomisation group. We estimated the difference between randomisation groups using a linear mixed model with practice included as a random effect. This would allow the estimation of potential differences in a full-scale trial. The intracluster correlation coefficient was estimated for each outcome, however great care should

be taken in the interpretation of these due to the small number of clusters.[27] All statistical analyses were undertaken using Stata V.15.

## Health economic evaluation

Data were collected on resource use from an NHS perspective to test data collection processes and to inform a future health economic evaluation estimating quality adjusted life years. A record was kept of resources required to provide GP training, as well as the length of initial and follow-up goal-setting consultations. Additional healthcare resource use was extracted from electronic health records by practices supported by a study researcher (EL) for the 6 months prior to randomisation and from randomisation to follow-up. Healthcare use was collected for: day-case and inpatient hospital admissions; outpatient visits; accident and emergency visits (A&E); consultations at the GP practice (GP, practice nurse, healthcare assistant, nurse practitioners); and other contacts, such as district nursing, allied health professional contacts, ambulance call outs and specialist nursing contacts.

Resource use was costed using the NHS reference costs[28] for secondary care and a published source for primary care contacts.[29] NHS reference costs were used to estimate a weighted average cost for day cases, non-elective short stay, non-elective long stay and elective admissions. For longer stays, additional days were costed using a weighted average of all excess bed day costs. For the first and second GP consultations in the goal-setting group, we had data on length of consultation and setting. The cost of providing training was estimated from a description given by the study researcher of duration and required staff. The cost of academic staff time was estimated using University pay scales (including employer's national insurance and superannuation payments). As the training would have relevance beyond the duration of the study, we estimated a useful life of 3 years and calculated an annual equivalent cost.[30] All costs are in 2015/2016 UK pounds sterling (£). As the duration of the study was 6 months, we did not discount costs and benefits. As the study size was very small with great variability in estimates of cost and effect, we did not estimate formal cost-effectiveness.

## Qualitative analysis

The video and audio recordings of control and goal-setting consultations were compared by the research team (CS, EL, AS, JM and Rebecca Harmston (RH)) to measure duration and explore the content and methodological implications for a future study. An in-depth analysis of the consultations using a conversation analytic informed approach[31] is reported elsewhere.[32]

A thematic framework-based analysis was used to analyse the focus groups recordings and transcripts[33] to assess the acceptability of the goal-setting intervention to patients and GPs and possible future improvements to the goal-setting intervention, training and trial design.

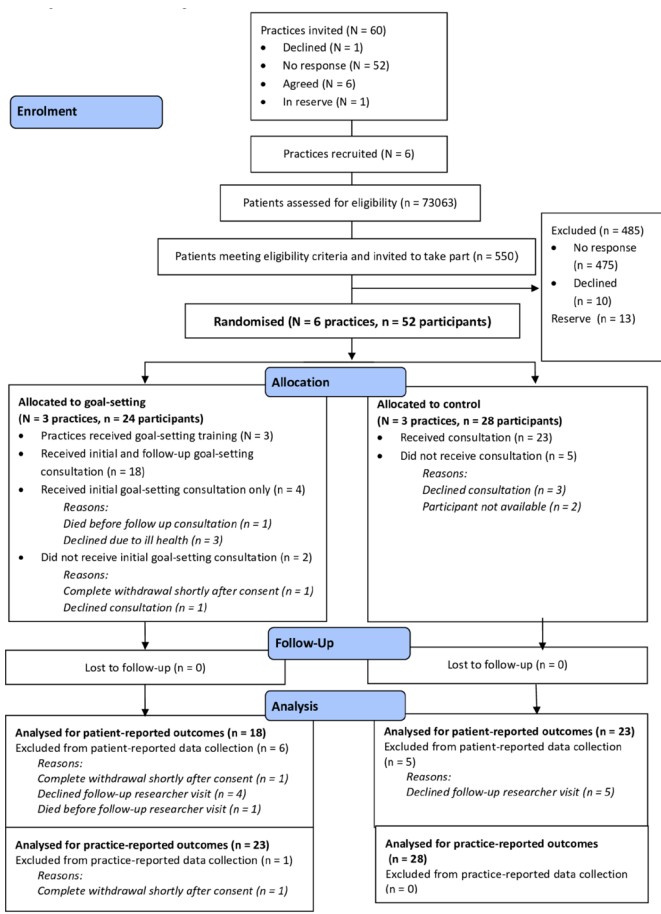

Figure 1  Consort flow diagram.

## Patient and public involvement (PPI)

Four individuals contributed to patient and public involvement (CG, RH, AM, Hillary Stringer (HS)). Two PPI representatives contributed to the design of the research as coapplicants on the initial application for funding (AM and HS) and steering group membership (AM and CG). PPI members contributed to the analysis and interpretation of the results, with one PPI representative reviewing and scoring video consultations using OPTION (RH) and a further two reviewing a selection of video consultation transcripts (AM and CG). Two PPI members reviewed and commented on the manuscript and are coauthors (AM and CG).

## RESULTS
### Recruitment and retention

Sixty general practices were invited with seven expressing interest and six being recruited (figure 1). Across the six practices (table 1), 550 patients met the eligibility criteria and were invited. In total, 52 patients were recruited with 24 belonging to practices randomised to goal-setting and 28 to practices in the control group. Thirteen patients were held in reserve from three practices which had recruited enough patients. The response rate was 12% ((52+13)/550). There was little variation in age, sex and

deprivation between those who participated and those who did not (online supplementary table 1). Two participants in the goal-setting group and five in the control group did not receive the initial consultation because they declined to attend, were unavailable or withdrew consent. Four participants in the goal-setting group did not receive the follow-up consultation because of ill health or death. Data collected directly from participants were available for 18 participants in the goal-setting group and 23 in the control group. Participant data collected from practices were available for 23 participants in the goal-setting group and 28 in the control group. Recruitment of practices took place between December 2016 and February 2017 and recruitment of patients between April and May 2017.

The control practices were in more urbanised areas with larger practice populations and more female GPs participating compared with goal-setting practices (table 1). The goal-setting group, compared with control (see table 2), had more patient participants who were female (54% compared with 29%), older (80 years old compared with 77), with a higher number of health problems (five compared with four) and medications (13.0 compared with 11.5), but similar quality of life. The control group had participants spread across all four IMD quartiles, whereas the goal-setting group had participants in only the second and third quartiles. All participants were white British and retired, except for one participant in the goal-setting group who was of working age but not employed and one in the control group who was self-employed. There was variation in participant baseline characteristics between practices in mean age (range 69.5–85.8 years old), proportion of females (range 25%–73%), number of medications (range 10.0–15.5) and number of health problems (range 3.0–7.5) across participating practices.

The mean initial consultation time in the goal-setting group was 23.0 min and in the control group was 19.2 min (table 3). GPs in the intervention group saw a mean of 4.4 patients (range 4–5), whereas GPs in the control group saw a mean of 3.8 patients (range 2–7). Patients spoke more in the goal-setting group initial consultation (mean GP:patient word count ratio (WCR) 1.35) than the control group (WCR 1.52), but this was not statistically significant. Dyadic OPTION scores for GPs perceptions of shared decision making were not statistically significantly higher in the goal-setting group compared with the control group, and CollaboRATE scores were similar. Observer OPTION scores showed large variation and inconsistency in scoring between the three research team members (data not presented).

Most patients set two or three goals (table 4) in the goal-setting intervention arm, with GPs and patients setting on average one more goal in practice 1 than in practice 3. The the most common types of goals were related to management of chronic conditions, walking, maintaining social and leisure interests and weight management (table 5). Forty-two of the 50 goals were

**Table 2** Baseline characteristics of patient participants

| Variable | | Control | Goal-setting |
|---|---|---|---|
| Number | | 28 | 24 |
| Female n (%) | | 11 (39) | 13 (54) |
| Age mean (SD) | | 77.18 (9.42) | 80.42 (8.72) |
| GPCOG category n (%) | Impairment and further investigations implied | 1 (4) | 0 (0) |
| | Informant interview required | 17 (61) | 19 (79) |
| | No cognitive impairment | 10 (36) | 5 (21) |
| Number of diagnoses* median (IQR) | | 4.00 (3.00, 5.00) | 5.00 (3.00, 6.00) |
| IMD national quartile n (%) | 1 | 5 (18) | 0 (0) |
| | 2 | 9 (32) | 14 (58) |
| | 3 | 3 (11) | 10 (42) |
| | 4 | 11 (39) | 0 (0) |
| Marital status n (%) | Divorced | 0 (0) | 2 (8) |
| | Living with partner | 0 (0) | 2 (8) |
| | Married | 12 (43) | 10 (42) |
| | Single | 2 (7) | 4 (17) |
| | Widowed | 14 (50) | 6 (25) |

*Based on Barnett list.[2]

GPCOG, General Practitioner assessment of Cognition; IMD, Index of Multiple Deprivation.

scored with a mean attainment score per patient of 1.45 (1=partially attained and 2=as expected) with 'partially attained' being the the most common outcome (table 4).

In the control arm, goals were rarely mentioned. Four usual-care GPs followed the care planning template recommended within the Avoiding Unplanned Admissions enhanced service,[13] one GP appeared to treat it as a normal problem-focused consultation and another GP focused solely on end of life issues.

### Outcome measures

As expected in this small feasibility study, there were no statistically significant differences between goal-setting and control from baseline to follow-up in PACIC score,

health-related quality of life as measured by EQ-5D-5L, number of medications or GPCOG score (table 6 which also shows the intraclass correlation coefficients). Capability as measured by ICECAP-O at 6 months, improved slightly more in the control group than in the goal-setting group, but the 95% CI includes zero (mean difference between groups −0.08, 95% CI −0.15 to −0.00).

There was considerable variation in healthcare use in the 6 months prior to randomisation and 6 months follow-up (table 7). Most healthcare contact increased in both the control and goal-setting groups, but district nurse contacts increased and inpatient admissions decreased only in the goal-setting group. Quality and

**Table 3** Characteristics of initial consultations

| | Intervention group | | | | Control group | | | | Mean difference between intervention and control (95% CI) |
|---|---|---|---|---|---|---|---|---|---|
| | Practice 1 (n=10) | Practice 2 (n=8) | Practice 3 (n=4) | Intervention total (n=22) | Practice 4 (n=7) | Practice 5 (n=9) | Practice 6 (n=7) | Control total (n=23) | |
| Duration of initial consultation (mins) mean (SD) | 24.1 (4.0) | 23.3 (4.4) | 19.9 (6.2) | 23.0 (4.6) | 14.3 (4.8) | 25.2 (5.7) | 16.3 (4.1) | 19.2 (6.9) | 3.88 (−3.25 to 11.01) |
| Dyadic OPTION scores mean (SD) | 65.3 (9.0) | 63.2 (6.4) | 62.5 (3.6) | 64.0 (7.2) | 63.5 (13.0) | 62.7 (4.0) | 42.1 (20.4) | 56.6 (16.2) | 7.57 (−6.37 to 21.50) |
| CollaboRATE scores mean (SD) | 7.8 (1.0) | 8.5 (0.9) | 8.8 (0.2) | 8.2 (1.0) | 7.0 (2.6) | 8.6 (0.7) | 8.7 (0.6) | 8.1 (1.8) | 0.20 (−1.06 to 1.47) |
| GP:patient word count ratio mean (SD) | 1.23 (0.40) | 1.41 (0.78) | 1.50 (1.05) | 1.35 (0.67) | 1.13 (0.45) | 1.92 (0.75) | 1.39 (0.52) | 1.52 (0.67) | −0.14 (−0.65 to 0.37) |

GP, general practitioner.

**Table 4**  Patient participants, goals set and attainment scores by practice

| | | Practice 1 | Practice 2 | Practice 3 | Overall |
|---|---|---|---|---|---|
| Number of patients | | 10 | 8 | 4 | 22 |
| Number of patients setting 1, 2 or 3 goals | 1 goal | 0 | 2 | 1 | 3 |
| | 2 goals | 3 | 4 | 3 | 10 |
| | 3 goals | 7 | 2 | 0 | 9 |
| Number of goals set | | 27 | 16 | 7 | 50 |
| Number of goals with data available for attainment scoring | | 21 | 15 | 6 | 42 |
| Number of goals in each attainment score category (category score) n (%) | Worse than expected (−1) | 1 (4.8) | 2 (13.3) | 1 (16.7) | 4 (9.5) |
| | No change (0) | 4 (19.0) | 0 (0.0) | 2 (33.3) | 6 (14.3) |
| | Partially attained (1) | 9 (42.9) | 5 (33.3) | 1 (16.7) | 15 (35.7) |
| | As expected (2) | 2 (9.5) | 3 (20.0) | 1 (16.7) | 6 (14.3) |
| | A little more (3) | 2 (9.5) | 4 (26.7) | 0 (0.0) | 6 (14.3) |
| | A lot more than expected (4) | 3 (14.3) | 1 (6.7) | 1 (16.7) | 5 (11.9) |
| Mean goal attainment score per patient (range −1 to 4) | | 1.43 | 1.67 | 1.0 | 1.45 |

Outcomes Framework data were collected at baseline and follow-up, but the results were uninformative due to low numbers and low variability (online supplementary table 2). There was one death in the goal-setting group due to cancer, which was judged to be unrelated to the intervention. The estimated cost of the goal-setting was £147 per patient, of which £95 related to costs of providing initial and follow-up GP consultations, and £43 related to the cost of GP training. There was a small cost for the study researcher to explain goal-setting. A mean cost of £50 per patient was incurred in the control group for the initial consultation. The single largest cost for the 6 months prior to recruitment and the 6 months of follow-up was inpatient stays (table 7). There were also substantial costs in other settings, for example in

**Table 5**  Categories of goals set

| Goal categories | Goals (n) |
|---|---|
| Management of chronic condition (non-medication) | 9 |
| Walking-related | 8 |
| Maintain interests | 5 |
| Management of chronic condition (medication-related) | 5 |
| Gain weight | 4 |
| Social participation | 3 |
| Healthy living | 3 |
| Balance/mobility | 3 |
| Gardening-related | 3 |
| Manual dexterity | 3 |
| Mental health | 2 |
| End of life management | 1 |
| Cooking/food preparation | 1 |
| Grand total | 50 |

general practice contacts and district nurse services. The types, number and associated costs of health service use varied considerably, as would be expected in a feasibility study.

### Acceptability

Eleven patients expressed interest in the focus group but only six were able to attend on the selected date. Two patients who were unable to attend took part in a telephone interview. Of the five GPs who delivered the intervention, four attended the focus group and one was unable to attend, so was interviewed face-to-face at the GP surgery. All six patient participants attending the focus group reported positive experiences and views of the intervention, particularly regarding the different emphasis of the consultation. Participants spoke of goal-setting providing clarity about what mattered to them, and helping them to plan and focus their lives.

> [Goal-setting] gives he or she a much better understanding of particularly what is worrying you, what your aims are, the things that you miss being able to do and to be able to actually explain it where [GPs] have time, because very often the GPs, you know, you've only got ten minutes. But with these consultations, you're actually able to talk to a doctor, as you would indeed a friend almost. (Patient 107)

Goal-setting appeared to function as a mechanism for helping make consultations patient-centred. This was reflected in the unanimous support for the intervention among the four GPs who attended the GP focus group and one GP who was interviewed by phone. GPs described the goal-setting consultations as '*more patient-centred*' and reflected on the consultation's '*therapeutic powers*' (GP10) compared with day-to-day general practice, which GPs felt could be dominated by '*box-ticking*' and '*target driven*' (GP018) medicine.

**Table 6** Change in outcome measures between groups at 6 months

| Variable | Control | | | | Intervention | | | | Difference, mean (SD) | Mean difference-in-difference between goal-setting and control (95% CI) | Intraclass correlation coefficient (95% CI) |
|---|---|---|---|---|---|---|---|---|---|---|---|
| | n | Baseline, mean (SD) | Follow-up, mean (SD) | Difference, mean (SD) | n | Baseline, mean (SD) | Follow-up, mean (SD) | Difference, mean (SD) | | | |
| Number of medication | 28 | 12.5 (8.19) | 12.79 (7.25) | 0.29 (2.65) | 23 | 13.61 (4.56) | 14.65 (4.44) | 1.04 (3.21) | | 0.76 (−0.85 to 2.37) | 0.00* |
| GPCOG | 23 | 7.35 (1.70) | 6.78 (2.19) | −0.57 (2.02) | 19 | 7.58 (1.30) | 7.00 (2.26) | −0.58 (2.29) | | 0.09 (−1.65 to 1.84) | 0.08 (0.00 to 0.77) |
| PACIC | 23 | 1.45 (0.30) | 1.85 (0.77) | 0.40 (0.69) | 18 | 1.94 (0.76) | 2.25 (0.70) | 0.31 (0.98) | | −0.09 (−0.60 to 0.42) | 0.00* |
| EQ-5D-5L | 23 | 0.54 (0.34) | 0.52 (0.35) | −0.02 (0.19) | 18 | 0.56 (0.25) | 0.55 (0.28) | −0.01 (0.15) | | 0.02 (−0.11 to 0.13) | 0.05 (0.00 to 0.94) |
| ICECAP-O | 22 | 0.72 (0.26) | 0.78 (0.20) | 0.06 (0.14) | 17 | 0.78 (0.12) | 0.77 (0.13) | −0.02 (0.06) | | −0.08 (−0.15 to 0.00) | 0.00* |

*The CI was not reported in cases when the ICC is zero as the SE is undefined in these cases.
EQ-5D-5L, 5 level EQ-5D; GPCOG, General Practitioner assessment of Cognition; ICECAP-O, ICEpop CAPability measure for Older people; PACIC, Patient Assessment of Chronic Illness Care.

**Table 7** Costs associated with healthcare use

| Resource use | 6 months prior to recruitment | | | | | | Recruitment to 6 month follow-up | | | | | |
|---|---|---|---|---|---|---|---|---|---|---|---|---|
| | Control | | | Goal-setting | | | Control | | | Goal-setting | | |
| | Total contacts n | Total cost £ | Mean cost £ (SD) | Total contacts n | Total cost £ | Mean cost £ (SD) | Total contacts n | Total cost £ | Mean cost £ (SD) | Total contacts n | Total cost £ | Mean cost £ (SD) |
| Community based services | | | | | | | | | | | | |
| GP | 157 | 4636 | 166 (164) | 89 | 2464 | 107 (115) | 177 | 5150 | 184 (150) | 124 | 4002 | 174 (145) |
| Other practice based | 97 | 922 | 33 (42) | 108 | 1080 | 47 (30) | 152 | 1823 | 65 (58) | 149 | 1529 | 66 (53) |
| District nurse | 148 | 3582 | 128 (546) | 198 | 6450 | 280 (1297) | 100 | 2879 | 103 (321) | 241 | 7450 | 324 (1384) |
| Other | 72 | 1434 | 51 (132) | 72 | 2601 | 113 (193) | 189 | 7652 | 273 (355) | 97 | 5510 | 240 (224) |
| All community based | 474 | 10575 | 378 (778) | 467 | 12594 | 548 (1520) | 618 | 15681 | 560 (719) | 611 | 16962 | 737 (1537) |
| Inpatient | 4 | 11291 | 403 (1113) | 16 | 28054 | 1220 (2584) | 12 | 35055 | 1252 (2203) | 13 | 39889 | 1734 (4815) |
| Outpatient | 45 | 4848 | 173 (208) | 51 | 7381 | 321 (397) | 41 | 4424 | 158 (202) | 52 | 6295 | 274 (329) |
| A&E | 1 | 138 | 5 (26) | 6 | 826 | 36 (74) | 15 | 2066 | 74 (109) | 16 | 2204 | 96 (128) |
| Total for all costs | | 26853 | 959 (1776) | | 48856 | 2124 (4031) | | 57226 | 2044 (2665) | | 65349 | 2841 (4968) |

A&E, Accident and Emergency; GP, general practitioner.

I felt almost as if I was trying to put on a different hat, you know, trying not to constantly interrupt them or to sort of sway them in any way, I was trying to give them the opportunity to just say what they wanted to say and set any goal that they wanted to and I, and it made me reflect on actually what I do during the day to day when I've got ten minutes with a patient and I'm very aware of the sort of pressure of, oh I've got to do a medication review and I've got to do this and oh no, their cholesterol's now 7 and oh gosh I've, have my colleagues already spoke to them about this and are they aware of X, Y and Z and actually it was quite nice in a way just take a step back and think, um I don't have to do that with this consultation, let's see what happens when the patient has more control over it. (GP025)

Patient participants spoke positively about the baseline researcher visit because it helped them understand the study and encouraged them to reflect on what was important. However, when discussing wider implementation across the health service, participants acknowledged that a home visit for each patient may be too costly and alternative provision would be acceptable to most people. Patients were reluctant to receive more paperwork as they felt that it was a burden for some people. When asked by the moderator to consider the acceptability of a group session to introduce people to the study and to the concept of goal-setting, all bar one of the patient participants at the focus group felt this would be acceptable.

Continuity of care was a concern for patient participants. While one person was disappointed not to see their own GP, three were positive about consulting with a different doctor, especially if it was difficult to see their usual GP. However, participants spoke of wanting more follow-up and consistency among the healthcare team in relation to their goals in the future; some participants felt there was a disconnection between the activity of goal-setting and their subsequent treatment by staff within the practice.

GPs stated that the experiential work, especially role play and skill spotting, was the most useful aspect of training. When discussing delivering training at scale, GPs felt e-training with opportunities to watch 'other people role-play', would fit in with their busy schedules. In addition, multiple shorter e-training modules, using a 'step-by-step' approach (GP014) that contributed to continuing professional development, would be attractive to GPs when implementing the intervention more widely.

## DISCUSSION

The process of setting goals in a GP consultation and follow-up over 6 months was acceptable to patients and unanimously supported by participating GPs. Recruitment and retention of practices and patients was achieved. A wide range of goals were set and, as expected with a feasibility study, there were no statistically significant differences in the main outcomes. Goal-setting consultations were a similar length to control consultations. The qualitative findings were that goal-setting helped patients and GPs focus on what was important and supported GPs to deliver more patient-centred care. Patient preparedness, continuity of care and being able to deliver training at scale were important considerations for future studies of goal-setting. Data on the number of health problems were not sufficiently robust for analysis because they were extracted from practice records using different processes. Asking GPs in the non-intervention group to undertake a video-recorded usual care planning consultation is likely to have altered practice compared with what would have happened within the enhanced service. An intention-to-treat analysis was undertaken to reduce the impact of protocol violations (eg, patients not receiving the prespecified intervention).

A Cochrane review, published in 2015, assessed the effects of personalised care planning (defined as goal-setting and action planning), for adults with long term health conditions compared with usual care.[8] While 19 randomised controlled trials (RCTs) were included, all except for one focused on single conditions. The one multiple condition study included patients who had high healthcare use and focused on care planning, with goal-setting as part of the process, across the wider healthcare system to reduce unplanned admissions.[34] The authors found an increase in quality of life (measured by SF36) in the intervention compared with control, however with 50% of participants lost to follow-up and intention to treat not undertaken, there is a possibility of a lost to follow-up bias in favour of the intervention. Our study has focused on goal-setting specifically in primary care.

A systematic review of randomised and non-randomised studies, published in 2017, looked at collaborative goal-setting or health priority setting for elderly people with a chronic condition or multimorbidity.[12] The authors found that in four of eight intervention studies, multifactorial approaches improved goal-setting or care planning, but the review did not assess health outcomes or quality of life. The authors concluded that future research was needed to determine the 'mix of essential elements within a multifactorial intervention to provide recommendations on daily practice'. Our study helps to answer this question by identifying some key requirements of goal-setting in primary care.

This was a feasibility study and the main implications are for the design of a subsequent definitive trial. Our objectives were to assess participant recruitment and retention, the acceptability of a goal-setting intervention to patients and GPs, the training needs of GPs, the content of control consultations, goal-setting and the feasibility of collecting relevant outcome measures.

We set out to recruit six practices, and seven (out of 60 invited) were willing to take part after one initial email invitation. Participant recruitment and retention was sufficient overall, but low in one practice (which

recruited four out of a target of ten). Reminder letters were not sent, but these may help all practices to recruit larger numbers if required in a future study. Seven participants, five from the control and two from goal-setting, did not receive the initial consultation because they declined the consultation, withdrew consent or were not able to attend. Possibly some were disappointed to be allocated to the control group.

Goal-setting was acceptable to participating patients and GPs, although they were a self-selecting group who were willing to take part in research into goal-setting. Goal-setting is unlikely to be relevant to everyone, but the positive response of participants in this feasibility study suggests that it is likely to have wider acceptability in general practice. Further research is needed to understand which patients will benefit most from goal-setting. The readiness of patients to undertake goal-setting appeared to be important. Although several goals were only partially attained, GPs and patients still felt them to be worthwhile, suggesting that the process of goal-setting has benefits, apart from the achievement of goals.

Training participating GPs in goal-setting was important, and participating GPs thought that the face-to-face training with role play used in the feasibility study could be replaced with online e-learning to allow delivery at scale to a wider GP workforce. The initial researcher visit was important to participants and the key elements of this visit would be delivered in a future trial using video and leaflet-based patient information aids, again to be developed using material collected during this feasibility study.

Goal-setting consultations were more focused on what matters to the patient than the control consultations. Key challenges in goal-setting included preparation and agreeing goals and we explore these further elsewhere.[32] Some patients were concerned that their goals were not considered in future consultations, which suggests that better communication of goals with the rest of the healthcare team will be needed. Planned follow-up of goals with the GP sooner than 6 months if needed would improve continuity of care, which is associated with lower mortality.[35]

We collected a wide range of outcome measures in order to assess their feasibility and suitability for use in a future trial. Both EQ-5D-5L and the ICECAP-O should be used in a future economic evaluation but would not be the best primary outcome measure for a trial of goal-setting. A recent study which aimed to improve the management of patients with multimorbidity, the 3D study, used the EQ-5D-5L as a primary outcome, but did not find any significant difference between arms.[36] It may be that the domains within the EQ-5D-5L are insensitive to changes in care for patients with multimorbidity and a measure of patient centred care such as PACIC is a more appropriate primary outcome measure as it contains a subscale to measure goal-setting. Baseline and follow-up data were collected during researcher visits, which could be replaced by postal questionnaires as the amount and complexity

of data to be collected would be reduced. Postal questionnaires are widely used in research and could either increase or reduce the completeness of follow-up data, depending on the preference of individuals for a visit rather than a postal form to complete.

Quality and Outcomes Framework data did not prove useful because of the small numbers and low variation. The observer OPTION scoring, initially developed within a rehabilitation context, had low consistency between researchers and therefore was not useful. A possible reason for this lack of consistency was that OPTION was developed for specific clinical decisions, and not for goal-setting which often involved multiple complex decisions.

Goal-setting can be valuable for GPs and patients seeking to agree the desired outcomes of care, particularly for older patients with multimorbidity. This study has demonstrated that it is acceptable and feasible in general practice, and a full trial is now needed to assess whether goal-setting improves important clinical outcomes for patients.

**Acknowledgements** We thank Rebecca Harmston for reviewing the video-consultations from the patient and public involvement perspective, Clara Yates and Gosia Majsak-Newman at the Norfolk and Suffolk Primary and Community Care Research Office for their support, and the patients and staff who took part in the study.

**Contributors** NS, JAF, CS and AS conceived the idea. All authors contributed to the design of the study. EL led the data collection which was supported by TW, IK and SJ. CS, JM and AS led the analysis of the qualitative data which was supported by SW. ABC undertook the statistical analysis. DT undertook the economic analysis. CG and AM provided a patient and public perspective and helped with qualitative analysis. AL led the partnership with NHS England and contributed to handling of adverse events. All authors contributed to the interpretation of the results. JAF drafted the initial manuscript. All authors revised the manuscript and approved the final version. NS is the guarantor.

**Funding** This paper presents independent research supported by the National Institute for Health Research (NIHR) under its Research for Patient Benefit (RfPB) Programme (Grant Reference Number PB-PG-0215-36079) and by the NIHR Collaboration for Leadership in Applied Health Research and Care East of England Programme. The views expressed are those of the author(s) and not necessarily those of the NHS, the NIHR or the Department of Health and Social Care. The funders did not have any role in the design, collection, analysis or interpretation of data or in writing the manuscript.

**Competing interests** None declared.

**Patient consent for publication** Not required.

**Ethics approval** Research ethics approval was obtained from the NHS Research Ethics Committee (16/EM/0411).

**Provenance and peer review** Not commissioned; externally peer reviewed.

**Data sharing statement** Dataset of quantitative data and statistical code is available from the corresponding author.

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
