## [Reviewer comments · BMJ Open]

ARTICLE DETAILS

TITLE (PROVISIONAL)	Can goal-setting for patients with multimorbidity improve outcomes in primary care?: cluster randomised feasibility trial
AUTHORS	Ford, John; Lenaghan, E; Salter, Charlotte; Turner, David; Shiner, A.; Clark, Allan; Murdoch, Jamie; Green, Carole; James, Sarah; Koopmans, Imogen; Lipp, Alistair; Moseley, Annie; Wade, Tom; Winterburn, Sandra; Steel, Nicholas

VERSION 1 - REVIEW

REVIEWER	Carol Sinnott University of Cambridge England.
REVIEW RETURNED	29-Jul-2018

GENERAL COMMENTS	This study assessed the feasibility of goal-setting for patients with multimorbidity with a view to undertaking a future definitive randomised controlled trial. Specific objectives were to assess 1) participant recruitment and retention2) the acceptability of a goal-setting intervention to patients and GPs,3) the training needs of GPs,4) the content of usual care planning consultations and goal-setting5) the feasibility of collecting relevant outcome measures While it appears to have achieved some of these aims, I found there were some silences on how this information would be used to inform the future trial. I did not see a justification for the sample size used in the study. How many practices did you intend to recruit? While a sample size calculation would not be expected, there is a need to explain that rationale for having three practices in each arm. There was no logic model or theory of mechanisms of action put forward for how the intervention was to effect change in the chosen outcomes. The acronyms for each of the outcomes (ICECAP, OPTION etc.) need to be explained and some information on what these outcomes are intending to measure is required. Did the baseline results compare favourably with other national cohorts? None of the stated objectives relate to generating preliminary evidence of the efficacy of the intervention, but significance levels at 5% are reported. There was also testing of association with multiple outcomes. There was no statement of the minimum clinically important difference for each of the outcomes. Perhaps using confidence intervals alone to infer the size and direction of treatment effect would be more appropriate.
--

	Regarding the ICCs used in the linear models, were these adopted from existing literature or were they calculated specifically for this trial? While blinding of GPs and patients was not possible, was there blinding of the research staff who collected outcome data or of the analysis team? What is "slight" cognitive impairment? How do the rates of cognitive impairment in your study sample compare with national rates (it seems high)? What was the intention of following up the GPCOG at follow-up? Explain how risk of unplanned admissions was determined; what was the 'Avoiding Unplanned Admissions' enhanced service and how did this service ascertain the risk? Were GPs paid for participating in the study (e.g. training workshops/searches/ consultations)? The training workshop encompassed shared-decision making skills and the Calgary Cambridge model- in my mind, these are different skills to that of helping patients set goals for their own care. Can you explain more about how GPs were trained to facilitate goal-setting in these sessions please? Once goals were set by the patient, did the GP support them in attaining these goals (i.e by referring to other providers/seeing them back for follow-up of medical intervention etc.)? Please clarify. Consultations were rated for shared decision making. How does this align with patient goal-setting? Is this a conventional/validated approach to measuring goal-setting? How were participants recruited for the focus groups? Only 6 attended but it looks like some also had a phone interview- did all get a chance to do a phone interview? Why was the scoring of OPTION scores so inconsistent between the research team that it rendered these results meaningless? Did they receive training /standard operating procedures for how to apply this tool? Page 9 line2- 8; this is a bit vague- what is the data to back up these findings? Was the increase in health care utilisation related to the setting of goals or care-planning? For instance, did GPs refer more in order to help patients attain goals/ deliver on care-plans? Page 9 line 33: "Costs were very heterogenous as would be expected" –this is not clear- do you mean they varied numerically or in terms of content? Page 10 line 15- what were the alternatives that patients suggested? Insufficient detail presented to assess the rigour of the qualitative analysis. For example, details on topic guide, interviewers, coders were not provided. Page 12 line 24 states that goal planning does not require any more time than standard care planning. Earlier you mention that the resources for care planning have been removed, so longer care planning consultations are no longer the norm. Can you clarify here if you are suggesting that it is possible to introduce goal planning (as delivered in this study) into routine consultations (which in the UK are ten minute consultations)? Table 5- was there any exploration in the focus groups of why patients' goals were not attained? The primary outcome did not appear to be sensitive to the intervention (albeit the study was presumably under-powered to show this). How will this influence the authors choice of outcome for the future trial? Do they expect that larger numbers will change
--	---

	this result? Will they proceed with this choice of outcome? I would recommend that the study findings are discussed in the light of the recent 3D study (Lancet June 2018) which showed no effect on HRQOL after an intensive, well-resourced intervention aiming to improve patient centred care. The results section of the abstract focuses on the statistical findings too much and not on the objectives on the study – suggest rewriting.
--	---

REVIEWER	Dr Manbinder Sidhu University of Leicester, UK
REVIEW RETURNED	20-Aug-2018

GENERAL COMMENTS	Overall, this is a very interesting paper exploring goal setting for patients with multi-morbidities in primary care. The paper is well-written; however, in places, leaves the reader desiring a greater depth of detail. The main point of contention is the lack of contextualisation of goal setting and what this might entail e.g. medication adherence, self-management, increasing physical activity, or increasing opportunities for behavioural change. This appears implicitly throughout the paper (and supplementary appendices/tables/figures). Article summary  - I believe there should be some comment about the study response rate, with regard to both patients and practices. - Also, the potential need for greater qualitative data collection to explore how goals are set using a patient-centred approach between GPs and patients Introduction  - Line 5- increasing global prevalence of LTCs? - Why is there a focus on primary care and GPs in particular? Why not consider practice nurses too? - Need to give greater context in which patients living with multi-morbidities would be expected to set goals towards e.g. e.g. medication adherence, self-management, increasing physical activity, or increasing opportunities for behavioural change. - Why it is important for patient's to set goals when living with multiple LTCs? Methods  - Given that GP practices were identified by the CRN, have they previously engaged in research trials? This may impact upon practice recruitment as part of the larger trial or why they showed a willingness to participate. Also, did they receive service support costs? - Recruitment- can you explicitly detail how you came to the decision of recruiting 10 patients per practice by sending 100 invitations via letter? Hence, why not send a letter to all who were eligible for each practice? Were reminder invitations sent to increase the response rate? - Where did the study researcher meet patients? Were patients given a PIL when informed consent was obtained? - Intervention- were senior consultation skills clinical by background? If yes, this needs to be stated in the manuscript. - How were the training needs of GPs, in order to set goals with patients, determined? I understand that you refer to Elwyn; but, was there any consultation with GPs prior to designing the training workshop? Some justification for the use of role play, as part of
--

training, would be welcomed (I am an advocate of using arts-based methods in health research!).

- A brief description of what the handbook contains would be useful. Was this a tool which GPs were expected to refer before or after consultations?
- Data- please provide numbers of how many consultations were video recorded and how many were audio-recorded. How patients attended the focus group and how many were interviewed over the phone?
- Both statistical analysis and health economic evaluation are well detailed. Input from PPI members is well-documented and increases rigour (thank you). Did PPI members also comment on the suitability of patient facing documentation (e.g. PIL) and/or the nature of/selection of questionnaires chosen to collect data?

Results

- Recruitment- The authors need to address the low response rate from practices expressing interest in the study, as well as patients. Although I am aware a 10% response rate is not entirely uncommon, this would be a greater challenge for a larger study, and authors may wish to consider alternative ways of engaging general practices.
- The final sentence of the first paragraph is incomplete.
- Baseline characteristics- in your discussion section, some explanation is needed to explain that men may have a greater preference for consultations with goal setting (compared to women). I believe this is the case with men taking part in studies using action planning in COPD (e.g. PSM-COPD trial).
- Also, there is no information on patient employment or ethnicity. Can this be included?
- Consultation- I think a key issue is whether goal setting leads to longer consultations, but given that goals were only 'partially attained' is the process worthwhile. I think some critique of using/completing the goal setting sheet within consultations is warranted.
- Was there any variation in the extent to which GPs were prepared for goal-setting in the consultation? One assumes it comes easier to some over others.
- Outcome measures- given the small numbers, statistically significant results were not expected. However, from reading the consort diagram, there were some significant issues following up patients to complete patient reported data collection. How was this completed- researcher visits can be costly and timely. Again, this challenge needs to be discussed more.
- Focus group data- the quotes from patients and those included for GPs do not match the interpretation provided. I would ask the authors to look again at this data and ask whether more appropriate quotes could be inserted and re-consider some of the claims made. I disagree that the first quote is related to focus of lives, but rather developing a shared understanding of the patient's illness with the GP and having the time and space to develop a communication style that is suited to both. The second quote does not necessarily exude support for the intervention, but rather a different style of patient centred consultation that includes goal setting, as opposed to a goal setting consultation per se. Hence, goal setting is being used as a proxy to develop better therapeutic alliances between GPs and patients living with multiple morbidities.

Discussion

- More discussion about the low response rate and how that may be curtailed in a larger trial.

	 - A stronger case needs to be put forward for the intervention, as at present, GPs and patients are in support of more patient-centred consultations with goal setting. - The nature of attrition and how it may be reduced needs further discussion. - Will HRQoL be the primary outcome in the main trial? If not, then which outcome will be? - Will you use the OPTION tool given its poor consistency among researchers? Is there an alternative that could be considered?
--	--

REVIEWER	Agnes Grudniewicz Telfer School of Management, University of Ottawa, Canada
REVIEW RETURNED	20-Sep-2018

GENERAL COMMENTS	The paper addresses an important gap in the literature on our understanding of goal setting in primary care, in particular its feasibility. I think this is a timely and important paper. However, prior to publication, the authors need to address a critical issue related to the conceptualization of “goal setting” and “decision making”. These are distinct processes with some, but not complete overlap. The two need to be conceptually differentiated and a justification needs to be given on why training and scoring of consultations was based on shared decision making. There is also a lack of clarity in level of analysis. Practices are randomized but GPs give the intervention. As such, I think it is important to specify the number of GPs participating within each practice (and the number not participating per practice), how many patients they each had (perhaps those delivering the intervention more than once would become more experienced), etc. I describe suggested revisions below: Major Comments Conceptual Issues  1. The definition of “personalized care planning” is given in the second paragraph of the introduction. However, the concept of goal setting could be better defined. The authors use the term “sharing” of goals – however, I am not sure this is accurate. I believe there is a level of agreement required in goal-setting that goes beyond sharing. Also, the definition includes only physicians, however, in most cases it is a care planner or nurse setting the goals. The paper could benefit from a stronger definition of goal setting given that this is the focus of the paper. 2. Given that the intervention (goal-setting) is compared to care planning (usual care), it would be valuable to conceptually tease-apart these two processes. The introduction focuses on goal setting as a part of care planning – in this case, it is difficult to understand why usual care is care planning. I think specifying more clearly the differences in the two processes will better support the paper. 3. The intervention training is based on shared decision making. However, this is conceptually a different process than goal-setting. They should be differentiated, both explained, and more detail provided on the training of the intervention group (the training seems to be on shared decision making not goal setting). Furthermore, this becomes problematic on page 6, line 35, where the authors state that consultations were scored based on a shared decision measure. This makes me feel that the intervention
---

was not a goal setting intervention but rather a shared decision making intervention. More justification needs to be provided on why the training & outcome measures were based on shared decision making.

Lack of Critical Details

4. The intervention is poorly described. Did all GPs in intervention practices receive training or just the ones with selected patients (and how many of them were there)? Why was the training built on shared decision-making? How do these two concepts differ? What is the Calgary Cambridge Guide (should be explained)? Is the training handbook available for readers to access? Does the usual care group (care planning) do any goal setting (much of care planning is goal-oriented)? How long was the training? Did everyone that was invited participate? Was there a measurement or assessment of skill?

a. The intervention description says that patients were given a face-to-face explanation of goal setting for 15 minutes. More detail should be provided here as I'm not sure what could be talked to patients about for 15 minutes (as written it comes across not very patient-centered).

b. On page 6, line 10, it says that agreed upon goals were documented. One of the biggest challenges in goal-setting is reaching agreement between the physician and the patient. How was this done? What happened when there was disagreement?

c. Does the 20 minute consultation include the 15 minutes with the researcher? How does the researcher's discussion with patients about their goal impact feasibility of goal setting in primary care (where there is normally no researcher to establish that conversation prior to the clinical consultation)? Is the discussion with the patient part of the intervention or more to introduce the study?

5. It is not clear how many participants were invited to an interview and how many participated in an interview. In the methods (page 6, lines 40-41), the authors imply that ALL participants unable to attend the focus group were interviewed by phone using the same topic guide. However, on page 9, we are informed there were 6 patient participants and 4 GP participants in the focus group, with only 1 GP interviewed by phone.

6. On page 10, the authors say that one person was disappointed not to see their own GP. This was not clear in the intervention description. Why were patients not seeing their own GPs? Was this the case for all participating patients? The authors should provide more detail earlier in the study on this.

Minor Comments

More Details Needed

1. On page 5, first paragraph, the authors state that there was a 6 month follow up. A brief note as to why a 6 month follow-up was chosen would be valuable for the readers.

2. Page 5, line 16, the authors mention that general practices were "recruited". Was this done by email, mail, personal contact?

3. On page 5, under recruitment, it says "practices" undertook a search of their patient registers. Was this done by physician or by practice? It may be worthwhile specifying whether the intervention is by practice or physician and whether all physicians in a practice were prepared to undertake the intervention if their patient was selected.

	4. On page 6, line 34, the authors say that all consultations were video or audio recorded. Was it both? When was it one over the other? 5. Page 8, line 20 – it looks like this paragraph was not completed. 6. In Table 3, can you provide a note for areas where the difference between the groups was statistically significant? 7. On page 8, lines 45-50, the authors discuss number of goals set. I am assuming that this is in the goal-setting group. Can the authors be more explicit that this paragraph is referring only to the intervention group? 8. Page 9, lines 31-33, the authors state “However, significant costs occurred outside the hospital setting, for example in general practice contracts and district nurse services.” This statement is rather vague and doesn’t provide much value. What do the authors mean by “significant”? Statistically significant? Is there an amount? 9. I think it is important to specify in the manuscript how many GPs participated in the intervention. Typos Page 4, line 13 is missing the word “on” Page 11, line 36 is missing the word “in”
--	---

REVIEWER	Alex McConnachie Robertson Centre for Biostatistics University of Glasgow Scotland
REVIEW RETURNED	07-Oct-2018

GENERAL COMMENTS	Ford et al report on a feasibility cluster RCT of a training intervention to implement goal setting for multimorbid patients. This review looks at the use of statistics in the paper. There is very little bad to say about this analysis. The flow of practices and patients through the study is clearly reported. The data are summarised clearly; perhaps Table 7 could also show the data at follow-up, as well as the changes from baseline, but that is a minor point. It is good to see the ICCs reported, even if they are not very informative – perhaps the results section could at least mention these. One thing about the study procedures – it is not clear whether the patient baseline questionnaires were administered before or after the practices were randomised. Ideally, the baseline data collection would take place first. Data about the initial care planning consultation has to be collected after randomisation, but the patient questionnaires could be done before. The flow diagram could indicate when these data were collected relative to practice randomisation. In the strengths and weaknesses section of the discussion, it is claimed that an ITT analysis protects against attrition bias. I’m not sure this is accurate – ITT is to do with analysing according to randomised group, regardless of whether the intervention is received. Missing data due to attrition is a separate issue. The first paragraph of the results section ends mid-sentence. Besides these minor points, I have nothing to add. I think this is a good report of a well conducted feasibility trial.
---

REVIEWER	Fiona Boland Royal College of Surgeons in Ireland, Dublin, Ireland.
REVIEW RETURNED	16-Oct-2018

GENERAL COMMENTS	This is a very interesting study examining the feasibility of goal setting for patients with multimorbidity in primary care. The manuscript is very clear but I have outlined a few specific comments and queries for clarification below: Methods: Eligibility – in the protocol more inclusion / exclusion criteria were given. It is stated that there were no changes from protocol so perhaps include more and/or refer to the protocol for more details (page 5, line 16). It would be useful also to include the link to the protocol in reference 13. Intervention – How many GPs from each practice were trained? (page 5, line 47) Results: Recruitment and retention - Final sentence in this section is incomplete (page, 8 line 19). Baseline characteristics of practices and participants – In the methods section of the manuscript (and in the protocol) it states that 100 patients were invited and patients were randomly selected within each IMD quartile, with extra from the least affluent quartile to increase participation. However, Table 1 indicates that more than 100 patients were invited in some practices and on page 8 line 7 it indicates that all eligible patients were invited, not a random sample? Please clarify if all eligible patients were recruited or a sample of 100 in those practices that identified more than 100 eligible. Was there a possible reason why there were no participants in the goal setting group from the lowest and highest IMD? Less eligible and hence invited in those quartiles? Consultation findings - Suggest referring to Table 5 at end of second paragraph (page 8, line 49). Outcome measures - Page 9, line 17: The result of the difference in ICECAP-O between the goal setting and usual care groups at six months reported here is different to that reported in Table 7? Please check all results. Discussion: The authors state that a goal setting intervention is feasible to deliver but it seems recruitment and response rate was very low. While mentioned in the limitations I think it could also be addressed in the implications for a future trial assessing effectiveness as it could be very difficult to recruit the required numbers (obviously depending on primary outcome and sample size). How might issues around recruitment in the feasibility trial be tackled in a future trial? Additionally, does IMD need to be considered in a future trial? Tables:
--

	Table 1 It would be nice to include % patients invited and % recruited for each practice also. Suggest removing practice level baseline characteristics (totals are also given in Table 3) or combining with Table 2, and renaming to patient characteristics. Need to check the 'practice level baseline characteristics' values also as some differ to the totals given in Table 3 (e.g. number of diagnoses). Table 7 Include 'at six months' in heading. All results in this table need to be double-checked and particularly ICECAP-O (compare to results in text – they differ). Abstract: Line 41: Change 'significantly' to 'significant' Spell out acronyms used.
--	---

VERSION 1 – AUTHOR RESPONSE

Reviewer	Ref	Section	Original Page	Original Line(s)	Comment	Response
1	1a	Paper	n/a	n/a	This study assessed the feasibility of goal-setting for patients with multimorbidity with a view to undertaking a future definitive randomised controlled trial. Specific objectives were to assess  1) participant recruitment and retention 2) the acceptability of a goal-setting intervention to patients and GPs, 3) the training needs of GPs, 4) the content of usual care planning consultations and goal-setting 5) the feasibility of collecting relevant outcome measures While it appears to have achieved some of these aims, I found there were some silences on how this information would be used to inform the future trial.	We have added and clarified information regarding recruitment and response rate for both patients and practices (see Recruitment and retention subsection: p9 para 1 line 4-5). We have also added more detail on the baseline characteristics and goal setting training delivered (see see Recruitment and retention subsection: p9 para 2 line 7-10 and Intervention subsection: p5 para 5 and p6 para 1 and 2). We have added a new section to the discussion setting out how this information would be used to inform the future trial (See Discussion p12 para 4 onwards).
1	1b	Paper	n/a	n/a	I did not see a justification for the sample size used in the study. How many practices did you intend to recruit? While a sample size calculation would not be expected, there is a need to	We intended to recruit 6 practices. The rationale for having three practices in each arm was to assess both the feasibility of the intervention and the content of the control consultations in several different practices, whilst keeping

Reviewer	Ref	Section	Original Page	Original Line(s)	Comment	Response
					explain that rationale for having three practices in each arm.	the number as low as possible to maximise efficiency.
1	2	Paper	n/a	n/a	There was no logic model or theory of mechanisms of action put forward for how the intervention was to effect change in the chosen outcomes.	We have revised the introduction to include how the intervention may improve the chosen outcomes. We did not have a formal logic model and have reported these qualitative results on mechanisms of action more fully in a separate paper (Salter C, Shiner A, Lenaghan E, et al. Setting goals with patients living with multimorbidity: qualitative analysis of general practice consultations. 2018. Manuscript submitted for publication).
1	3	Paper	n/a	n/a	The acronyms for each of the outcomes (ICECAP, OPTION etc.) need to be explained and some information on what these outcomes are intending to measure is required. Did the baseline results compare favourably with other national cohorts?	We have spelled out the ICECAP-O acronym in the abstract and main text. We have added a description of the five attributes of the ICECAP-O measure to the methods section and a reference is provided for further information “capability (as measured through the five attributes of attachment, security, role, enjoyment and control in the ICEpop CAPability measure for older people questionnaire (ICECAP-O) [19]) (ICEPOP is the name of the UK MRC-funded programme through which the index was developed)” (see Data and statistical analysis subsection: p6, para 5, lines 3-6). OPTION is not an acronym. In the methods section we state that OPTION assesses shared decision making and provide references for further reading “GPs and patient participants were asked to complete an assessment of shared decision making during each consultation using the CollaboRATE scale [23] for patients and dyadic OPTION scale [24]” (see Data and statistical analysis subsection: p6, para 4, lines 1-2).
1	4	Paper	n/a	n/a	None of the stated objectives relate to generating preliminary evidence of the efficacy of the intervention, but significance levels at 5% are reported. There was also testing of association	Our aim was to “assess the feasibility of goal-setting for patients with multimorbidity, ..., with a view to undertaking a future definitive randomised controlled trial” (Introduction: p4, para 4, lines

Reviewer	Ref	Section	Original Page	Original Line(s)	Comment	Response
					with multiple outcomes. There was no statement of the minimum clinically important difference for each of the outcomes. Perhaps using confidence intervals alone to infer the size and direction of treatment effect would be more appropriate.	6-8). An analysis of the outcomes was needed to inform the sample size calculation for the definitive trial and ensure feasibility of the analysis. We do not report minimally clinically important differences as we were not seeking to judge the effectiveness of the intervention. We have removed the p values and only reported confidence intervals.
1	5	Statistical analysis	6	53-54	Regarding the ICCs used in the linear models, were these adopted from existing literature or were they calculated specifically for this trial?	The ICCs reported in Table 6 were calculated specifically from the trial data
1	6	Methods: randomisation	5	42	While blinding of GPs and patients was not possible, was there blinding of the research staff who collected outcome data or of the analysis team?	Practice level data was provided by practice staff, so it was not possible for this group to be blinded because practices were aware of the allocation. Patient level questionnaire data was collected through self-completion, therefore there is less chance of outcome assessor bias. The statistician that completed the quantitative data analysis was blinded to group. Blinding of research staff was not possible in this small scale study as these staff, by necessity, were also involved in liaising with practices to organise the intervention. We have added further details (see Methods: p5, para 4, lines 3-6).
1	7	Abstract	2	39	What is "slight" cognitive impairment? How do the rates of cognitive impairment in your study sample compare with national rates (it seems high)? What was the intention of following up the GPCOG at follow-up?	We have replaced "slight cognitive impairment" with 'overall 28% of patient participants had no cognitive impairment' in the abstract. The GPCOG was used to screen for cognitive impairment and we did not compare our impairment rates compare with national rates for 2 reasons: 1. only 52 patients were included in this study which was not designed to be nationally representative, and 2. national data on cognitive impairment for this particular population (adults, at risk of unplanned admission, diagnosed with ≥ 2 chronic health problems) is not available. GPCOG was included as part of a range of questionnaires collected at baseline and follow-up, but it is unlikely that this will be collected at

Reviewer	Ref	Section	Original Page	Original Line(s)	Comment	Response
						follow-up during the definitive study.
1	8	Abstract Methods: eligibility criteria	2 5	21 18	Explain how risk of unplanned admissions was determined; what was the 'Avoiding Unplanned Admissions' enhanced service and how did this service ascertain the risk?	We have added further details to the methods "To be eligible, practices had to be using risk stratification to identify patients at high risk of unplanned admission (for example by participating in the Avoiding Unplanned Admissions Enhanced Service [13])." (Methods: p5, para 2, lines 3-4) According to the enhanced service to reduced avoidable admissions for those at highest risk "The practice will use an appropriate risk stratification tool or alternative method, if a tool is not available, to identify vulnerable older people, high risk patients and patients needing end-of-life care who are at risk of unplanned admission to hospital." (see https://www.england.nhs.uk/wp-content/uploads/2014/08/avoid-unplanned-admissions.pdf)
1	9	Paper	n/a	n/a	Were GPs paid for participating in the study (e.g. training workshops/searches/consultations)?	Yes, practices were reimbursed for their time and travel expenses. A statement has been added to the methods section "Practices were reimbursed for staff time and travel to undertake the research and deliver the intervention." (see p5, Methods: para 2, lines 6-7)
1	10	Methods: intervention	5	47-55	The training workshop encompassed shared-decision making skills and the Calgary Cambridge model- in my mind, these are different skills to that of helping patients set goals for their own care. Can you explain more about how GPs were trained to facilitate goal-setting in these sessions please?	We have added more details about the explicit training on goal setting, and compared goal setting and shared decision making (see Intervention subsection: p5, para 5, also p4, Introduction: para 2, lines 9-14).
1	11	Paper	n/a	n/a	Once goals were set by the patient, did the GP support them in attaining these goals (i.e by referring to other providers/seeing them back for follow-up of medical intervention etc.)? Please clarify.	The goals were set by the GP and patient in collaboration and the GP supported the patient to achieve the goals as required. Examples include: a GP giving a patient a leaflet about a local social group for bereaved people; a GP suggesting a patient contact the local Age Concern group about finding a driving companion; advice to contact Citizens Advice about writing a formal will etc. We have added more details (see

Reviewer	Ref	Section	Original Page	Original Line(s)	Comment	Response
						Intervention subsection: p6, para 3, lines 3-5).
1	12	Abstract: results Methods: intervention	2 5	40 50	Consultations were rated for shared decision making. How does this align with patient goal-setting? Is this a conventional/validated approach to measuring goal-setting?	We have clarified the definition of goal setting in the introduction ("the sharing of realistic goals by doctors and patients and agreement of the best course of action" Introduction: p4, para 2, lines 2-4). Shared decision making is a key part of goal setting, but differs in that shared decision making is concerned with specific clinical decisions, whereas goal setting is concerned with the patient's priorities. Goal attainment was measured using the conventional goal attainment scoring scale (see Table 4) which can only be used in the intervention group. Goal setting may have other effects, such as improving shared decision making and quality of life. Therefore we collected data on shared decision making, in addition to other outcomes in all participants.
1	13	Methods: data	6	36-40	How were participants recruited for the focus groups? Only 6 attended but it looks like some also had a phone interview- did all get a chance to do a phone interview?	More detail has been added to the methods section and results section about the recruitment to the focus group and telephone interview. Methods section – "All patients in the intervention group were sent a letter of invitation to the focus group, except two who indicated at the researcher visit they did not want to take part." (see Data and statistical analysis subsection: p7, para 1, lines 5-6) Results section – "Eleven patients expressed interested in the focus group but only six were able to attend on the selected date. Two patients who were unable to attend agreed to a telephone interview." (see Acceptability subsection: p10, para 4, lines 1-2)
1	14	Results: Consultation findings Discussion:	8 11	42-43 43-45	Why was the scoring of OPTION scores so inconsistent between the research team that it rendered these results meaningless? Did they receive training /standard operating	The Observer OPTION Manual was read by all those undertaking the scoring (see http://www.glynelwyn.com/uploads/2/4/0/4/24040341/-observeroption5manual_july_13_2016.docx.pdf). However the main

Reviewer	Ref	Section	Original Page	Original Line(s)	Comment	Response
		implications for a definitive trial			procedures for how to apply this tool?	difficulty was that OPTION scoring is well suited to situations with individual specific clinical decisions, but not for goal setting consultations which involved multiple decisions, some of which were not clearly clinical. It is likely that some heterogeneity of scoring occurred as a consequence of this, as reviewers had to provide an overall score for multiple different decisions and negotiations. We concluded that OPTION is not feasible for use in assessing goal setting, and will not be using it in a future trial.. We have added a comment to the discussion (see Discussion, p13, para 5, lines 2-5line 930-3).
1	15	Results: Consultation findings	9	2-8	Page 9 line2- 8; this is a bit vague- what is the data to back up these findings?	We have removed this section as the qualitative results are given in a separate paper (submitted) and there is not space in this paper to adequately present the qualitative data on these findings.
1	16	Paper	n/a	n/a	Was the increase in health care utilisation related to the setting of goals or care-planning? For instance, did GPs refer more in order to help patients attain goals/ deliver on care-plans?	The study was not powered to draw conclusions about differences in health care utilisations or the underlying factors.
1	17	Results: outcome measures	9	33	Page 9 line 33: "Costs were very heterogeneous as would be expected" –this is not clear- do you mean they varied numerically or in terms of content?	We have clarified to state "The types, number and associated costs of health service use varied considerably, as would be expected in a study with a comparatively small sample size." (Outcome measures subsection: p10, para 3, lines 13-14)
1	18	Results: acceptability...	10	15-	Page 10 line 15- what were the alternatives that patients suggested?	Agreed, we have added text to clarify (Acceptability subsection: p11 para 2 line 6-9)
1	19	Discussion: implications for clinical practice	12	24	Page 12 line 24 states that goal planning does not require any more time than standard care planning. Earlier you mention that the resources for care planning have been removed, so longer care planning consultations are no longer the norm. Can you clarify here if you are suggesting that it is possible to introduce goal planning (as	Our intervention group used a 20 minute initial consultation and we found that even though the control group was only being reimbursed for a 10 minute appointment the control group spent almost as long as the goal setting group. Therefore we feel that it would be possible to introduce goal setting within a standard consultation. Our goal setting consultations took

Reviewer	Ref	Section	Original Page	Original Line(s)	Comment	Response
					delivered in this study) into routine consultations (which in the UK are ten minute consultations)?	more than 10 minutes, but so did the controls, so we think that in routine practice it is likely to be the case that these patients with complex conditions are rarely in and out of the consultation in 10 minutes, and that goal setting probably takes a similar time to a routine consultation, for this patient group.
1	20	Table 5	21	n/a	Table 5- was there any exploration in the focus groups of why patients' goals were not attained?	The purpose of this paper was to report the main feasibility study aims and objectives, rather than provide an in-depth analysis of the qualitative data, which would require a long and unwieldy paper. An in-depth analysis of the qualitative data has been undertaken and has been submitted for publication (submitted to BJGP 2018).
1	21	Discussion: implications for a definitive trial	11	n/a	The primary outcome did not appear to be sensitive to the intervention (albeit the study was presumably under-powered to show this). How will this influence the authors' choice of outcome for the future trial? Do they expect that larger numbers will change this result? Will they proceed with this choice of outcome? I would recommend that the study findings are discussed in the light of the recent 3D study (Lancet June 2018) which showed no effect on HRQOL after an intensive, well-resourced intervention aiming to improve patient centred care.	The study did not have a primary outcome and was not powered to detect differences between outcomes. We are aware of the 3D study, which was published after submission, and our future trial is more likely to use a patient centred outcome measure, such as PACIC. We have added a sentence to the discussion. "A recent study which aimed to improve the management of patients with multimorbidity, the 3D study, used the EQ5D5L as a primary outcome, but did not find any significant difference between arms (Salisbury 2018). It may be that the domains within the EQ5D5L are insensitive to changes in care for patients with multimorbidity and a measure of patient centred care such as PACIC is a more appropriate primary outcome measure as it contains a sub scale to measure goal setting." (See Discussion: p13, para 9, lines 3-8)
1	22	Abstract	2	37-46	The results section of the abstract focuses on the statistical findings too much and not on the objectives on the study – suggest rewriting.	We agree and have removed the following text from the abstract "The goal-setting group had higher scores for shared decision-making compared to the usual care planning group, but not statistically significantly. There was no significant difference in EQ5D or PACIC between groups, and ICECAP-O was slightly higher in

Reviewer	Ref	Section	Original Page	Original Line(s)	Comment	Response
						usual care planning.” We have added text relating to the length of consultation and examples of goals set.
2	23	Paper	n/a	n/a	The main point of contention is the lack of contextualisation of goal setting and what this might entail e.g. medication adherence, self-management, increasing physical activity, or increasing opportunities for behavioural change. This appears implicitly throughout the paper (and supplementary appendices/tables/figures).	We have revised the introduction to include more contextualisation and examples of what goal setting may entail.
2	24	Article summary	3	n/a	I believe there should be some comment about the study response rate, with regard to both patients and practices. Also, the potential need for greater qualitative data collection to explore how goals are set using a patient-centred approach between GPs and patients	We have revised the summary to include statements on response rate and also on qualitative data.
2	26	Introduction	4	5	Line 5- increasing global prevalence of LTCs?	We have added a reference to the global Burden of Disease Programme to support this.
2	27	Introduction	4	n/a	Why is there a focus on primary care and GPs in particular? Why not consider practice nurses too?	Goal setting is relevant to both primary and secondary care, but this particular study is focussed on primary care. Goal setting for patients with multimorbidity is likely to be most effective when established during a GP-patient consultation because it is the GP and patient who are the primary decision makers in terms of management of chronic disease. Undoubtedly practice nurses and other allied health professionals have an important role in helping patients attain their goals, but we feel that agreement of goals between GP and patient is key. We have included a statement of this in the methods (Intervention subsection: p6 para 2 line 11-14)
2	28	Introduction	4	n/a	Need to give greater context in which patients living with multimorbidities would be expected to set goals towards e.g. e.g. medication adherence, self-management, increasing physical activity, or increasing opportunities for behavioural change.	We have revised the introduction to include more context as to why goal setting is important and some examples (see Introduction: p4, para 2).

Reviewer	Ref	Section	Original Page	Original Line(s)	Comment	Response
					Why it is important for patient's to set goals when living with multiple LTCs?	
2	30	Methods	5	n/a	Given that GP practices were identified by the CRN, have they previously engaged in research trials? This may impact upon practice recruitment as part of the larger trial or why they showed a willingness to participate. Also, did they receive service support costs?	Those practices recruited are considered "research active" and most will have taken part in trials previously. Recruitment of practices through the CRN is a standard for NIHR funded research. We don't think it will have much impact on a larger study because in a definitive trial the practices would be recruited via other regional CRNs. The practices received reimbursement for their time and travel expenses. This has been included in the methods section. "Practices were reimbursed for staff time and travel to undertake the research and deliver the intervention." (Methods: p5, para 2, lines 6-7)
2	31	Methods	5	27-35	Recruitment- can you explicitly detail how you came to the decision of recruiting 10 patients per practice by sending 100 invitations via letter? Hence, why not send a letter to all who were eligible for each practice? Were reminder invitations sent to increase the response rate?	We estimated that 10 patients per practice would be sufficient to answer the feasibility study questions. Based on an expected response rate of 10%, that led to 100 invitations being proposed. In reality the number of patient eligible in each practice ranged from 47 to 124 and therefore all were invited. We have clarified this in the methods section, see Methods: p5, para 3, line 4. Reminders were not sent because this was not in the protocol or part of the funding. However we would consider this in the definitive study.
2	32	Methods	6	4-5	Where did the study researcher meet patients? Were patients given a PIL when informed consent was obtained?	Participants were visited at home and a PIL was sent with the letter of invitation. These details have been added to the methods section, see Methods: p5, para 3, line 3 and lines 6-7.
2	33	Methods	5	47-54	Intervention- were senior consultation skills clinical by background? If yes, this needs to be stated in the manuscript.	Two were not clinical, but have substantial experience in teaching and designing communication skills sessions for undergraduates and postgraduates. One was a GP (as stated).
2	34	Methods	5	47-54	How were the training needs of GPs, in order to set goals with patients, determined? I understand that you refer to Elwyn; but, was there any	We have added more details to the section on training to explain more about what aspects of the training was specific to goal setting and further details of the role play.

Reviewer	Ref	Section	Original Page	Original Line(s)	Comment	Response
					consultation with GPs prior to designing the training workshop? Some justification for the use of role play, as part of training, would be welcomed (I am an advocate of using arts-based methods in health research!).	
2	35	Methods	5	52	A brief description of what the handbook contains would be useful. Was this a tool which GPs were expected to refer before or after consultations?	We have included a description of the handbook in the methods section. "The handbook contained information about the study and a "how to" guide for goal-setting, including theoretical background and examples of goal setting. See Intervention subsection: p5, para 5, lines 12-14.
2	36	Methods: data			Data- please provide numbers of how many consultations were video recorded and how many were audio-recorded.	41 consultations were video-recorded and 4 were audio-recorded. These details have been included in the manuscript (see Data and statistical analysis subsection: p6, para 5, line 1).
2	37	Methods			How [many] patients attended the focus group and how many were interviewed over the phone?	Details have been added to the manuscript, see Acceptability subsection: p10, para 4, lines 1-2. "Eleven patients expressed interest in the focus group but only six were able to attend on the selected date. Two patients who were unable to attend took part in a telephone interview."
2	38	Methods	7	48-54	Both statistical analysis and health economic evaluation are well detailed. Input from PPI members is well-documented and increases rigour (thank you). Did PPI members also comment on the suitability of patient facing documentation (e.g. PIL) and/or the nature of/selection of questionnaires chosen to collect data?	Two PPI members were part of the steering group, contributed to the design of the study and are co-authors of this paper. They also reviewed the participant information sheet and letter of invitation.
2	39	Results: recruitment and retention	8	n/a	Recruitment- The authors need to address the low response rate from practices expressing interest in the study, as well as patients. Although I am aware a 10% response rate is not entirely uncommon, this would be a greater challenge for a larger study, and authors may wish to consider alternative ways of engaging general practices.	We have noticed an error in the response rate. There were thirteen patients who were on a reserve list. Therefore, the response rate was 12%. In a future study we would consider reminder letters to improve recruitment. We have included text in the discussion "Reminder letters were not sent, and these may help all practices to recruit larger numbers if required in a future study." (see Discussion: p12, para 5, lines 3-4).

Reviewer	Ref	Section	Original Page	Original Line(s)	Comment	Response
						Practices were invited by a two emails from the Clinical Research Network to research active practices. Since the target number of practices were recruited further engagement to improve practice recruitment was not undertaken.
2	40	Results: recruitment and retention	8	20	The final sentence of the first paragraph is incomplete.	Apologies, this was supposed to contain the recruitment dates, which have now been added (see Results: p9, para 1, lines 12-14).
2	41	Results: baseline characteristics...	8	n/a	Baseline characteristics- in your discussion section, some explanation is needed to explain that men may have a greater preference for consultations with goal setting (compared to women). I believe this is the case with men taking part in studies using action planning in COPD (e.g. PSM-COPD trial).	Our data do not show that that men are substantially more likely than women to take part in goal setting. 28 of 52 patients were male (54%). We are not aware of published literature suggesting that men have a greater preference for goal setting compared to women. In the PSM-COPD trial 63% of participants were male, but it is difficult to draw conclusions about gender preferences for goal setting from this study.
2	42	Results: baseline characteristics...	8	n/a	Also, there is no information on patient employment or ethnicity. Can this be included?	We have added details to the results section. "All participants were white British and retired, except for one participant in the goal-setting group who was of working age but not employed and one in the usual care planning group who was self-employed." See Results: p9, para 2, lines 7-8.
2	43	Results: consultation findings	8	n/a	Consultation- I think a key issue is whether goal setting leads to longer consultations, but given that goals were only 'partially attained' is the process worthwhile. I think some critique of using/completing the goal setting sheet within consultations is warranted.	As part of the intervention we purposefully provided a longer consultation length to allow sufficient time. Our analysis suggested that even partially attained goals were still seen as worthwhile by patients. No patients or GP specifically commented on the goal setting sheet in the focus groups, but analysis of the consultation data suggested that the patient held goal setting sheet helped prepare patients for the consultation. We have added more to the results and discussion.
2	44	Results	n/a	n/a	Was there any variation in the extent to which GPs were prepared for goal-setting in the consultation? One assumes it comes easier to some over others.	We have undertaken a more in-depth analysis of the qualitative data to explore preparedness of GPs, presented in a paper (submitted 2018)

Reviewer	Ref	Section	Original Page	Original Line(s)	Comment	Response
2	45	Results: outcome measures	9	n/a	Outcome measures- given the small numbers, statistically significant results were not expected. However, from reading the consort diagram, there were some significant issues following up patients to complete patient reported data collection. How was this completed- researcher visits can be costly and timely. Again, this challenge needs to be discussed more.	Follow-up data was collected through a researcher visit to a participant's home, as stated in the methods section "Data collected from patients during a researcher visit at baseline and six months". We accept that this may not be possible in a large scale trial and have added this to the discussion.
2	46	Results: acceptability & implications	9-10	n/a	Focus group data- the quotes from patients and those included for GPs do not match the interpretation provided. I would ask the authors to look again at this data and ask whether more appropriate quotes could be inserted and re-consider some of the claims made. I disagree that the first quote is related to focus of lives, but rather developing a shared understanding of the patient's illness with the GP and having the time and space to develop a communication style that is suited to both. The second quote does not necessarily exude support for the intervention, but rather a different style of patient centred consultation that includes goal setting, as opposed to a goal setting consultation per se. Hence, goal setting is being used as a proxy to develop better therapeutic alliances between GPs and patients living with multiple morbidities.	The quote illustrates a point of clarity that the person had reached regarding their goals. Goal setting required them to focus on what really mattered to them. This was not necessarily anything to do with an illness the person had. We have provided a number of examples of such goals (e.g. walking, meeting a partner). The quote demonstrates that the consultation with the GP provided this focus. we have emphasised this point further (Acceptability subsection: p10 para 4 line 6-7) We agree with the reviewer on the second point, that goal setting can help to develop a better therapeutic alliance. This is the point being made by the GP who "described the goal-setting consultations as more patient-centred and reflected on its 'therapeutic powers'". We would argue that patient-centred consultations do not just 'occur', but will necessarily include components that embody mechanisms that promote a patient centred consultation. A key finding from the study is that framing the consultation as one in which goal-setting is the primary activity functions in this way. The reported support from all GPs for the intervention is rooted in this perspective and the quote provides more detail of how the GP worked hard to deliver a patient-centred consultation through his interaction with the patient. However, we accept this could be clearer and have emphasised this point (see

Reviewer	Ref	Section	Original Page	Original Line(s)	Comment	Response
						Acceptability subsection: p10, para 6, lines 3-6).
2	47	Discussion	11-12		More discussion about the low response rate and how that may be curtailed in a larger trial.	We have noticed an error in the response rate. There were thirteen patients who were on a reserve list. Therefore, the response rate was 12%. In a future study we would consider reminder letters to improve recruitment. We have included text in the discussion "Reminder letters were not sent, and these may help all practices to recruit larger numbers if required in a future study." (see Discussion: p12, para 5, lines 3-4).
2	48	Introduction	11-12		A stronger case needs to be put forward for the intervention, as at present, GPs and patients are in support of more patient-centred consultations with goal setting.	We have revised the introduction to improve the justification for the intervention.
2	49	Discussion	11-12		The nature of attrition and how it may be reduced needs further discussion.	Attrition is now discussed in the discussion section.
2	50	Discussion	11-12		Will HRQoL be the primary outcome in the main trial? If not, then which outcome will be?	We do not expect quality of life to be the primary outcome in the main trial and have added discussion of this in the discussion section.
2	51	Discussion	11-12		Will you use the OPTION tool given its poor consistency among researchers? Is there an alternative that could be considered?	No, we are not planning on using OPTION in the definitive study. Instead we intend to use PACIC which measures patient centredness.
3	52	Paper	n/a	n/a	The paper addresses an important gap in the literature on our understanding of goal setting in primary care, in particular its feasibility. I think this is a timely and important paper. However, prior to publication, the authors need to address a critical issue related to the conceptualization of "goal setting" and "decision making". These are distinct processes with some, but not complete overlap. The two need to be conceptually differentiated and a justification needs to be given on why training and scoring of consultations was based on shared decision making.	We believe that goal setting is a means to deliver shared decision making. We agree that they are linked but conceptually different and so have revised the introduction to highlight the difference.

Reviewer	Ref	Section	Original Page	Original Line(s)	Comment	Response
3	53	Paper	n/a	n/a	There is also a lack of clarity in level of analysis. Practices are randomized but GPs give the intervention. As such, I think it is important to specify the number of GPs participating within each practice (and the number not participating per practice), how many patients they each had (perhaps those delivering the intervention more than once would become more experienced), etc.	Table 1 provides the characteristics of participating GPs, such as experience and gender, partnership status and fulltime or part time. GPs in the intervention group saw a mean number of 4.4 patients (range 4 to 5), whereas GPs in the control group saw a mean of 3.8 patients (range 2 to 7). We have added this to the results section (see Recruitment and retention subsection, p9, para 3, line 3). We do not have robust data on the number of GPs not participating during the study.
3	54	Paper	n/a	n/a	Conceptual issues (1 of 3): The definition of “personalized care planning” is given in the second paragraph of the introduction. However, the concept of goal setting could be better defined. The authors use the term “sharing” of goals – however, I am not sure this is accurate. I believe there is a level of agreement required in goal-setting that goes beyond sharing. Also, the definition includes only physicians, however, in most cases it is a care planner or nurse setting the goals. The paper could benefit from a stronger definition of goal setting given that this is the focus of the paper.	We have clarified the definition of goal setting in the context of care planning for patients with multimorbidity. The definition is now “Goal-setting is the sharing of realistic goals by health professionals and patients and agreement of the best course of action”.
3	55	Paper	n/a	n/a	Conceptual issues (2 of 3): Given that the intervention (goal-setting) is compared to care planning (usual care), it would be valuable to conceptually tease-apart these two processes. The introduction focuses on goal setting as a part of care planning – in this case, it is difficult to understand why usual care is care planning. I think specifying more clearly the differences in the two processes will better support the paper.	We have revised the introduction to highlight the problems with current care planning and why usual care planning does not necessarily mean goal setting.
3	56	Paper	n/a	n/a	Conceptual issues (3 of 3): The intervention training is based on shared decision making. However, this is conceptually a different process than goal-	We have substantially revised the introduction to tease apart the difference between shared decision making and goal setting. Essentially we believe that goal

Reviewer	Ref	Section	Original Page	Original Line(s)	Comment	Response
		Methods: data	6	35	setting. They should be differentiated, both explained, and more detail provided on the training of the intervention group (the training seems to be on shared decision making not goal setting). Furthermore, this becomes problematic on page 6, line 35, where the authors state that consultations were scored based on a shared decision measure. This makes me feel that the intervention was not a goal setting intervention but rather a shared decision making intervention. More justification needs to be provided on why the training & outcome measures were based on shared decision making.	setting embodies shared decision making. We assessed the impact of the intervention with a range of outcomes, of which shared decision making was one, and the others included goal attainment, patient centeredness, capability and quality of life.
3	57	Methods: intervention	5	n/a	Lack of critical details - The intervention is poorly described. Did all GPs in intervention practices receive training or just the ones with selected patients (and how many of them were there)? Why was the training built on shared decision-making? How do these two concepts differ? What is the Calgary Cambridge Guide (should be explained)? Is the training handbook available for readers to access? Does the usual care group (care planning) do any goal setting (much of care planning is goal-oriented)? How long was the training? Did everyone that was invited participate? Was there a measurement or assessment of skill?	We have added more detail to the intervention. Table 1 provides details of the GPs who took part. We have added details of the contents of the training manual (see Intervention subsection: p6, para 5, lines 12-16). Goal-setting consultations were only held with GPs who had been trained, even if s/he was not their usual clinician and this has been added (see Intervention subsection: p6, para 2, lines 1-2). We have added more detail to the intervention section. We would prefer not to make the handbook available to readers yet, because we are revising it for use in the definitive trial. The results state "In the control arm, goals were rarely mentioned" (Recruitment and retention subsection: p9, para 5, line 1). The methods state that the training involved a "three hour experiential workshop" (see Intervention subsection: p5, para 5, lines 1-2). One GP (practice 3) was able to attend the training but was not able to deliver the training for personal reasons. We have added this to

Reviewer	Ref	Section	Original Page	Original Line(s)	Comment	Response
						the methods (see Intervention subsection: p5, para 5, lines 3-4). An informal pre-test and post-test evaluation of the training was undertaken, but because of the small numbers (n=5) it was not sufficiently robust to present. However, it did demonstrate that confidence in goal-setting increased and that the role-play was the most valued aspect of the training (CS)
3	58	Methods: intervention	5	n/a	The intervention description says that patients were given a face-to-face explanation of goal setting for 15 minutes. More detail should be provided here as I'm not sure what could be talked to patients about for 15 minutes (as written it comes across not very patient-centered).	The associated goal-setting sheet was also discussed with the participants. We have revised this sentence to improve clarity (see Intervention subsection: p6, para 2, lines 1-4).
3	59	Methods: intervention	6	10	On page 6, line 10, it says that agreed upon goals were documented. One of the biggest challenges in goal-setting is reaching agreement between the physician and the patient. How was this done? What happened when there was disagreement?	A more in-depth analysis of the qualitative data has been undertaken to explore some of the enablers and barriers to achieving agreement and this is presented in a submitted paper (submitted 2018).
3	60	Methods: intervention	6	n/a	Does the 20 minute consultation include the 15 minutes with the researcher? How does the researcher's discussion with patients about their goal impact feasibility of goal setting in primary care (where there is normally no researcher to establish that conversation prior to the clinical consultation)? Is the discussion with the patient part of the intervention or more to introduce the study?	No. 15 minutes was spent with a researcher during the baseline visit. 20 minutes was then spent with the GP. We have revised the sentence to improve clarity. We initially planned that the researcher visit would be simply to provide information. However during a more in-depth analysis of the qualitative data became clear that the preparedness of patients was important to effective goal-setting and this may be influenced by the researcher visit. we have included a paragraph on the role of the researcher visit and added text to the discussion. See "Patient participants spoke positively about the baseline researcher visit because it helped them understand the study and encouraged them to reflect on what was important. However, when discussing wider

Reviewer	Ref	Section	Original Page	Original Line(s)	Comment	Response
						implementation across the health service, participants acknowledged that a home visit for each patient may be too costly and alternative provision would be acceptable to most people." (Acceptability subsection: p11, para 2, lines 1-5)
3	61	Methods: data	6	n/a	It is not clear how many participants were invited to an interview and how many participated in an interview. In the methods (page 6, lines 40-41), the authors imply that ALL participants unable to attend the focus group were interviewed by phone using the same topic guide. However, on page 9, we are informed there were 6 patient participants and 4 GP participants in the focus group, with only 1 GP interviewed by phone.	We have clarified the methods section and the results section. See "All patients in the intervention group were sent a letter of invitation to the focus group, except two who indicated at the researcher visit they did not want to take part....Patient or GP participants unable to attend the focus groups were interviewed by phone or face-to-face using the same topic guide." (Data and statistical analysis subsection: p7, para 1, lines 2-5). And "Eleven patients expressed interest in the focus group but only six were able to attend on the selected date. Two patients who were unable to attend took part in a telephone interview. Of the five GPs who delivered the intervention, four attended the focus group and one was unable to attend, so was interviewed face-to-face." (Acceptability subsection: p10, para 4, lines 1-4)
3	62	Results: acceptability & implications	10	18-19	On page 10, the authors say that one person was disappointed not to see their own GP. This was not clear in the intervention description. Why were patients not seeing their own GPs? Was this the case for all participating patients? The authors should provide more detail earlier in the study on this.	Participants were required to see one of the GPs who had received the training. We have clarified the methods to make this clear. See Intervention subsection: p6, para 2, lines 9-10.
3	63	Methods: setting	5	8	Minor comments / more details needed (1 of 9): On page 5, first paragraph, the authors state that there was a 6 month follow up. A brief note as to why a 6 month follow-up was chosen would be valuable for the readers.	We have added a sentence "Six months was long enough for patients and GPs to work towards the agreed goals, but not so long that the goals would have been forgotten." (See Methods: p5, para 1, lines 2-4)

Reviewer	Ref	Section	Original Page	Original Line(s)	Comment	Response
3	64	Methods: eligibility criteria	5	16	Minor comments / more details needed (2 of 9): Page 5, line 16, the authors mention that general practices were "recruited". Was this done by email, mail, personal contact?	It was done by email. These details have been added to the methods section (see Methods: p5, para 2, lines 1-2).
3	65	Methods: recruitment	5	27	Minor comments / more details needed (3 of 9): On page 5, under recruitment, it says "practices" undertook a search of their patient registers. Was this done by physician or by practice? It may be worthwhile specifying whether the intervention is by practice or physician and whether all physicians in a practice were prepared to undertake the intervention if their patient was selected.	The searches were undertaken by a practice administrator and a clinician checked the list. This has been added to the methods section (see Methods: p5, para 3, lines 1-2). The intervention was allocated by practice and delivered by 1-2 GPs in each practice. The methods state "The Norwich Clinical Trials Unit independently randomised three practices to goal-setting and three to control" (see Methods: p5, para 4, lines 1-2). Participating patients were required to see one of the GPs who had been trained and the methods have been clarified to include this (see Intervention subsection: p6, para 2, lines 9-10).
3	66	Methods: data	6	34	Minor comments / more details needed (4 of 9): On page 6, line 34, the authors say that all consultations were video or audio recorded. Was it both? When was it one over the other?	41 consultations were video-recorded and 4 were audio-recorded. These details have been included in the manuscript (see Data and statistical analysis subsection: p6, para 6, line 1).
3	67	Results: recruitment and retention	8	20	Minor comments / more details needed (5 of 9): Page 8, line 20 – it looks like this paragraph was not completed.	Apologies, this was supposed to contain the recruitment dates, which have now been added (see Recruitment and retention subsection: p9, para 1, lines 12-14).
3	68	Table 3	19	n/a	Minor comments / more details needed (6 of 9): In Table 3, can you provide a note for areas where the difference between the groups was statistically significant?	The consort statement is clear that this analysis should not be done (item 15 of consort statement: "Unfortunately significance tests of baseline differences are still common; they were reported in half of 50 RCTs trials published in leading general journals in 1997. Such significance tests assess the probability that observed baseline differences could have occurred by chance; however, we already know that any differences are caused by chance. Tests of baseline differences are not necessarily wrong, just illogical. Such hypothesis testing is superfluous

Reviewer	Ref	Section	Original Page	Original Line(s)	Comment	Response
						and can mislead investigators and their readers. Rather, comparisons at baseline should be based on consideration of the prognostic strength of the variables measured and the size of any chance imbalances that have occurred.”
3	69	Results: consultation findings	8	45-50	Minor comments / more details needed (7 of 9): On page 8, lines 45-50, the authors discuss number of goals set. I am assuming that this is in the goal-setting group. Can the authors be more explicit that this paragraph is referring only to the intervention group?	Yes and the text has been amended.
3	70	Results: outcome measures	9	31-33	Minor comments / more details needed (8 of 9): Page 9, lines 31-33, the authors state “However, significant costs occurred outside the hospital setting, for example in general practice contracts and district nurse services.” This statement is rather vague and doesn’t provide much value. What do the authors mean by “significant”? Statistically significant? Is there an amount?	Agreed. We have changed the word significant to substantial to remove any suggestion that this is based on statistical significance.
3	71	Paper	n/a	n/a	Minor comments / more details needed (9 of 9): I think it is important to specify in the manuscript how many GPs participated in the intervention.	Six GPs were trained, but one withdrew before the intervention was delivered for personal reasons. Table 1 shows the GPs who took part and a sentence has been added to the methods about the GP who withdrew. “One GP attended the training but withdrew prior to delivering the intervention for personal reasons” (see Intervention subsection: p5, para 5, lines 3-4).
3	72	Introduction	4	13	Page 4, line 13 is missing the word “on”	Agreed. Text amended.
3	73	Discussion: implications for a definitive trial	11	36	Page 11, line 36 is missing the word “in”	Agreed. Text amended.
4	74	Table 7	23	n/a	This review looks at the use of statistics in the paper. There is very little bad to say about this analysis. The flow of practices	We agree that presenting the data at follow-up is possible, but our outcome measure was the change in the outcome rather than the

Reviewer	Ref	Section	Original Page	Original Line(s)	Comment	Response
					and patients through the study is clearly reported. The data are summarised clearly; perhaps Table 7 could also show the data at follow-up, as well as the changes from baseline, but that is a minor point.	outcome itself, so we would prefer so not report this as the temptation would then be to add another analysis comparing the outcome at follow-up.
4	75	Results	8-10	n/a	It is good to see the ICCs reported, even if they are not very informative – perhaps the results section could at least mention these.	We have added this to the results section.
4	76	Consort diagram	16	n/a	One thing about the study procedures – it is not clear whether the patient baseline questionnaires were administered before or after the practices were randomised. Ideally, the baseline data collection would take place first. Data about the initial care planning consultation has to be collected after randomisation, but the patient questionnaires could be done before. The flow diagram could indicate when these data were collected relative to practice randomisation.	Practice-level baseline data (ie. GP and practice characteristics) were collected prior to practices being randomised. Due to project timescales, it was decided that randomisation would occur after patients had expressed interest (a minimum of 10 per practice except for practice 3 where only 4 Eols were received) but prior to the researcher visits at which consent was taken – therefore all of the patient-level baseline data were collected after randomisation. We have added this clarification to the methods.
4	77	Discussion: strengths & weaknesses	11	22-24	In the strengths and weaknesses section of the discussion, it is claimed that an ITT analysis protects against attrition bias. I'm not sure this is accurate – ITT is to do with analysing according to randomised group, regardless of whether the intervention is received. Missing data due to attrition is a separate issue.	We agree the ITT does not protect against bias per-se. However, our ITT analysis reported here was in relation to the number of medications for which we had only 1 person missing (who withdrew consent for data collection), this included participants who withdrew and did not respond to the questionnaire data. We have removed reference to attrition bias.
4	78	Results	8	20	The first paragraph of the results section ends mid-sentence. Besides these minor points, I have nothing to add. I think this is a good report of a well conducted feasibility trial.	Apologies, this was supposed to contain the recruitment dates, which have now been added (see Recruitment and retention subsection: p9, para 1, lines 12-14).
5	79	Methods: eligibility criteria	5	16-23	This is a very interesting study examining the feasibility of goal setting for patients with multimorbidity in primary care. The manuscript is very clear but I have outlined a few specific comments and queries for clarification below:	We have added the inclusion and exclusion criteria of the participating GP practices. See “To be eligible, practices had to be using risk stratification to identify patients at high risk of unplanned admission (for example by participating in the Avoiding

Reviewer	Ref	Section	Original Page	Original Line(s)	Comment	Response
					Eligibility – in the protocol more inclusion / exclusion criteria were given. It is stated that there were no changes from protocol so perhaps include more and/or refer to the protocol for more details (page 5, line 16). It would be useful also to include the link to the protocol in reference 13.	Unplanned Admissions Enhanced Service [13]), have at least one Good Clinical Practice trained GP and nurse, be available to attend the goal-setting training and not be a single handed practice” (see Methods: p5, para 2, lines 2-6). We have included the web link to the protocol in the references.
5	80	Methods: intervention	5	47-55	Intervention – How many GPs from each practice were trained? (page 5, line 47)	Six GPs were trained, but one withdrew before the intervention was delivered for personal reasons. Table 1 shows the GPs who took part and a sentence has been added to the methods about the GP who withdrew. “One GP attended the training but withdrew prior to delivering the intervention due to personal reasons” (see Intervention subsection: p5, para 5, lines 3-4).
5	81	Results	8	20	Recruitment and retention - Final sentence in this section is incomplete (page, 8 line 19).	Apologies, this was supposed to contain the recruitment dates, which have now been added (see Recruitment and retention subsection: p9, para 1, lines 12-14).
5	82	Methods: recruitment	5	27-35	Baseline characteristics of practices and participants – In the methods section of the manuscript (and in the protocol) it states that 100 patients were invited and patients were randomly selected within each IMD quartile, with extra from the least affluent quartile to increase participation. However, Table 1 indicates that more than 100 patients were invited in some practices and on page 8 line 7 it indicates that all eligible patients were invited, not a random sample? Please clarify if all eligible patients were recruited or a sample of 100 in those practices that identified more than 100 eligible. Was there a possible reason why there were no participants in the goal setting group from the lowest and highest IMD? Less eligible and hence invited in those quartiles?	We anticipated that in there would be a larger number of patients eligible and some kind of sampling would be required. In reality the number of patient eligible in each practice ranged from 47 to 124 and therefore all were invited. We have clarified this in the methods section, see Methods: p5, para 3, line 4.
5	83	Results: consultation findings	8	49	Consultation findings - Suggest referring to Table 5 at end of second paragraph (page 8, line 49).	We have added a reference to the table (renamed Table 4) at the end of this paragraph (see Recruitment

Reviewer	Ref	Section	Original Page	Original Line(s)	Comment	Response
						and retention subsection: p9, para 4, line 6).
5	84	Results: outcome measures	9	17	Outcome measures - Page 9, line 17: The result of the difference in ICECAP-O between the goal setting and usual care groups at six months reported here is different to that reported in Table 7? Please check all results.	The results in the table are correct and we have updated the results section.
5	85	Discussion	11-12	n/a	The authors state that a goal setting intervention is feasible to deliver but it seems recruitment and response rate was very low. While mentioned in the limitations I think it could also be addressed in the implications for a future trial assessing effectiveness as it could be very difficult to recruit the required numbers (obviously depending on primary outcome and sample size). How might issues around recruitment in the feasibility trial be tackled in a future trial? Additionally, does IMD need to be considered in a future trial?	We have noticed an error in the response rate. There were thirteen patients who were on a reserve list. Therefore, the response rate was 12%. In a future study we would consider reminder letters to improve recruitment. We have included text in the discussion "Reminder letters were not sent, and these may help all practices to recruit larger numbers if required in a future study" (see Discussion: p12, para 5, lines 3-). In a future study, we don't think that IMD is likely to add much.
5	86	Table 1	17	n/a	Table 1: It would be nice to include % patients invited and % recruited for each practice also.	Agreed. We have included % in the table.
5	87	Tables 1-3	17-19	n/a	Suggest removing practice level baseline characteristics (totals are also given in Table 3) or combining with Table 2, and renaming to patient characteristics.	We have removed this data.
5	88	Table 1	17	n/a	Need to check the 'practice level baseline characteristics' values also as some differ to the totals given in Table 3 (e.g. number of diagnoses).	Thank you. We've checked the data and ensured that the upper CI for number of diagnoses in the Goal-setting group is correctly reported as 6.00.
5	89	Table 7	23	n/a	Table 7: Include 'at six months' in heading. All results in this table need to be double-checked and particularly ICECAP-O (compare to results in text – they differ).	Agreed. Title amended. We have checked the results to ensure they are correct. On reflection the ICC=0.00 are probably more accurately described as "not estimated"
5	90	Abstract	2	41	Abstract: Line 41: Change 'significantly' to 'significant' Spell out acronyms used.	We have removed "significantly" from the abstract and spelled out the acronyms.

VERSION 2 – REVIEW

REVIEWER	Dr Manbinder Sidhu Health Services Management Centre, University of Birmingham, UK
REVIEW RETURNED	14-Dec-2018

GENERAL COMMENTS	I believe the authors have addressed reviewers comments appropriately and offered necessary clarification where possible. The paper is now much improved, clearer, with focused discussion with regard to goal setting. In particular, how such an intervention may be delivered in a real world setting (both from a NHS provider and patient perspective). I have one further, very minor comment. Methods (Pg 5, line 6)- the authors provide their reasoning for a six months follow up as; however, could this statement be supported by evidence that confirms six months is enough for patients and GPs to work towards agreed goals? I have no further comments.
---

REVIEWER	Agnes Grudniewicz Telfer School of Management, University of Ottawa Canada
REVIEW RETURNED	15-Jan-2019

GENERAL COMMENTS	The authors have substantially increased the clarity of manuscript and done a good job addressing reviewer comments. There remain a few areas that could benefit from further detail. The authors have done a nice job with the abstract. It's nice and clear. The authors expanded on the issue of goal setting versus shared decision making in the introduction. However, this area could use some further polish, which should be quite easy to do. Specifically: 1. Page 4, line 16: The authors provide a definition of goal setting based on a systematic review of care planning by Coulter and colleagues. I suggest that the authors reframe this to say: "For the purposes of this study, we define goal setting as...". Goal setting for people with chronic conditions does not yet have a single accepted definition (for example, the idea of needing to set realistic goals has been debated) and by stating that this is the definition that is being adopted for the study, it leaves room for a discussion of what goal setting for this population really means.2. Page 4, line 22: The authors say "Despite the recommendation" – this statement is quite far from the recommendation and hence is a bit unclear. I would suggest the authors restate the recommendation they are referring to3. Page 4, line 27: The authors write "both involve partnership working, choices, options and decisions". This is unclear as I am not sure what "partnership working" means. I think that the paper would benefit from a more clear argument as to why goal setting should include shared decision making and I think this is important to do well.4. Page 4, lines 32 to 40: I remain confused about the distinction between care planning and the goal setting intervention (perhaps because I am not in the UK where care planning is considered usual care). The authors provide a definition of care planning
---

which explicitly states goal setting (lines 33 to 34). I think the authors need to acknowledge this and provide an explicit statement as to why the goal setting intervention is different from usual care (aside from the training) if care planning already includes goal setting. Is this because in practice care planning does not actually include goals? Is it the degree to which goal setting is used in usual care planning? This will also help the reader understand lines 41-44 where you state that the recommendation is that patients in the top 2% have a care plan (which is the usual care). An explicit statement clarifying the distinction between usual care (care plans, which by your definition include goal setting) and goal setting would solve this issue.

5. Page 9, line 58: The authors discuss a “care planning template” for usual care. Are goals listed on this template and GPs did not follow that section? Important to note.

The authors did an excellent job better describing the intervention in this version of the manuscript. Only one issue remains unclear – how did the authors get from participating practices to participating GPs? Were all GPs in the practice invited to participate and it was based on self-selection? Was there a limit to how many GPs participate per practice? Can the authors add a line clarifying this? I think it is important to understand (even if it is not possible to know what percentage of GPs per practice agreed to participate). Some more minor comments:

1. Page 6, line 14: Authors state 3 questions to consider. I counted 5 (albeit in 3 bullets).
2. Page 8, line 6: “... was used to analyse the focus groups...” – that should say “focus group TRANSCRIPTS”.
3. Page 8, line 12: This paragraph would be more clear if it started with a statement of how many PPIs were involved overall and then go into their roles in the project.
4. Page 10, lines 4-5: Not clear if this means for all their patients? So the GP that treated it as end of life issues did this for all 3-4 patients they saw as part of the trial?
5. Page 13, line 22: The authors write “were around preparation and agreeing goals”. Is this missing the word “on”?
6. Page 13, line 49: “patients seeking to agree the desired outcomes of care”. Is this missing the word “to”?

The manuscript could benefit from some changes to the tables for added clarity.

Table 1: Can authors add the number of GPs per practice in total? Also, the row of “characteristics of participating GPs” is quite hard to read. Not clear if that means that for practice one it was 2 male GPs, both were partners and both worked part time? Maybe just another way of noting that would make it easier for the reader to understand quickly. And lastly, for the row of “patients assessed for eligibility, n” – the Control numbers are unclear. What is “108(0.6)”? It doesn’t conform to the row title.

Table 4: The title of this table is “Number of goals set and goal attainment score” but that makes it hard to understand the rows titled “Number of goals per patient” – are those numbers by practice the number of people that set those goals? If yes – can this be retitled somehow to add clarity. Otherwise, a reader skimming this table will struggle to understand. Also, can the authors add under “means core of goal attainment per person” what that score is out of (i.e., highest and lowest score)? It would increase readability to have that directly in the table.

REVIEWER	Alex McConnachie Robertson Centre for Biostatistics, University of Glasgow, Scotland, UK
REVIEW RETURNED	23-Dec-2018

GENERAL COMMENTS	I am happy with the authors' responses to my previous comments. I have just a couple of very minor points. Some explanation could be given as to why some of the ICCs are not (cannot?) be reported in Table 6. It is stated in the results that patients spoke more in the goal-setting group, but that other consultation measures "were not statistically significantly higher". However, looking at Table 3, the estimated between-group difference in WCR has a CI that includes zero.
--

REVIEWER	Fiona Boland Royal College of Surgeons in Ireland, Ireland
REVIEW RETURNED	24-Dec-2018

GENERAL COMMENTS	As far as I can see the authors have sufficiently addressed the previously comments. I have a few minor follow-on comments: It is stated in the paper that "six month follow-up was long enough for patients and GPs to work towards goals...." Was this the authors opinion or from somewhere else? Data and Statistical analysis section, page 6 (second line in section): "Data were collected from patients....."Needs minor rephrasing Data and Statistical analysis section, page 7 (second last para in section): "Key characteristics were compared using a linear mixed model with practice as a random effect". This appears to be in relation to baseline characteristics (not recommended by consort statement) – please clarify. Data and Statistical analysis section: what statistical package did you use for the analysis? Table 6: (1) Need to state that the mean and SD are reported in the table. (2) ICC: "Not estimated" rows, should this be 0.00? Typically the ICC is 0.00 when the between-subject variation is very small compared to within-subject variation.
---

VERSION 2 – AUTHOR RESPONSE

Reviewer	Ref	Section	Original Page	Original Line(s)	Comment	Response
2	91	Paper	n/a	n/a	I believe the authors have addressed reviewers comments appropriately and offered necessary clarification where possible. The paper is now much improved, clearer,	Thank you.

Reviewer	Ref	Section	Original Page	Original Line(s)	Comment	Response
					with focused discussion with regard to goal setting. In particular, how such an intervention may be delivered in a real world setting (both from a NHS provider and patient perspective).	
2	92	Methods	5	6	I have one further, very minor comment. Methods (Pg 5, line 6)- the authors provide their reasoning for a six months follow up as; however, could this statement be supported by evidence that confirms six months is enough for patients and GPs to work towards agreed goals?	This was primarily a pragmatic decision as there is no robust evidence on the most appropriate length of time. Other studies (such as the 3D study) have also used 6 months as a review period for goals. One of our findings was that participants may have benefit with follow-up earlier than 6 months, for example a 3 month telephone conversation. In the discussion we state "Planned follow-up of goals with the GP sooner than six months if needed would also improve continuity of care".
3	93	Introduction	4	10-11	The authors have substantially increased the clarity of manuscript and done a good job addressing reviewer comments. There remain a few areas that could benefit from further detail. The authors have done a nice job with the abstract. It's nice and clear. The authors expanded on the issue of goal setting versus shared decision making in the introduction. However, this area could use some further polish, which should be quite easy to do. Specifically: 1. Page 4, line 16: The authors provide a definition of goal setting based on a systematic review of care planning by Coulter and colleagues. I suggest that the authors reframe this to say: "For the purposes of this study, we define goal setting as...". Goal setting for people with chronic conditions does not yet have a single accepted definition (for example, the idea of needing to set realistic goals has been debated) and	Agreed. We have edited the text to read "For the purposes of this study, we define care planning as 'a conversation in which patients and clinicians agree on goals and actions for managing the patient's conditions'"

Reviewer	Ref	Section	Original Page	Original Line(s)	Comment	Response
					by stating that this is the definition that is being adopted for the study, it leaves room for a discussion of what goal setting for this population really means.	
3	94	Introduction	4	15	2. Page 4, line 22: The authors say “Despite the recommendation” – this statement is quite far from the recommendation and hence is a bit unclear. I would suggest the authors restate the recommendation they are referring to.	Agreed. We have added the following text to the introduction “Despite the recommendation that health professionals should establish patient goals with individuals with multimorbidity,”
3	95	Introduction	4	19	3. Page 4, line 27: The authors write “both involve partnership working, choices, options and decisions”. This is unclear as I am not sure what “partnership working” means. I think that the paper would benefit from a more clear argument as to why goal setting should include shared decision making and I think this is important to do well.	We have amended the text in the introduction to make the argument for including shared decision making in goal setting stronger. The edited text is “The goal setting approach is more likely to be effective if it incorporates shared decision making, the process by which health professionals and patients make decisions together based on the best available evidence [11], because the goals and actions agreed will be more patient-centred leading to greater engagement in the process by patients.”
3	96	Introduction	4	Para 3	4. Page 4, lines 32 to 40: I remain confused about the distinction between care planning and the goal setting intervention (perhaps because I am not in the UK where care planning is considered usual care). The authors provide a definition of care planning which explicitly states goal setting (lines 33 to 34). I think the authors need to acknowledge this and provide an explicit statement as to why the goal setting intervention is different from usual care (aside from the training) if care planning already includes goal setting. Is this because in practice care planning does not actually include goals? Is it the degree to which goal setting is used in usual care	We have clarified these two sections of text in the introduction. Firstly, we have amended the text to highlight that goal setting is rarely an important element of the care planning process in the UK by including the following text “Goal-setting should be, but is rarely, an important element of the care planning process in the UK”. Secondly, we have expanded the description of usual care in the aims section of the introduction by adding the following description “compared to control consultations (the usual care planning process undertaken in UK primary care which rarely includes goal setting”.

Reviewer	Ref	Section	Original Page	Original Line(s)	Comment	Response
					planning? This will also help the reader understand lines 41-44 where you state that the recommendation is that patients in the top 2% have a care plan (which is the usual care). An explicit statement clarifying the distinction between usual care (care plans, which by your definition include goal setting) and goal setting would solve this issue.	
3	97	Results	9	44	5. Page 9, line 58: The authors discuss a “care planning template” for usual care. Are goals listed on this template and GPs did not follow that section? Important to note. The authors did an excellent job better describing the intervention in this version of the manuscript. Only one issue remains unclear – how did the authors get from participating practices to participating GPs? Were all GPs in the practice invited to participate and it was based on self-selection? Was there a limit to how many GPs participate per practice? Can the authors add a line clarifying this? I think it is important to understand (even if it is not possible to know what percentage of GPs per practice agreed to participate).	Goal setting is not mentioned in the national template. We have added a statement to the methods to clarify “This may have involved a national care planning template, which does not include goal setting, from the Avoiding unplanned admissions enhanced service: proactive case finding and patient review for vulnerable people [13].” To be included practices had to nominate 2 GPs who would be able to attend the training and deliver goal setting consultations or deliver control consultations. In one practice, one of the nominated GPs was not able to attend the training, but this is likely to have minimal impact because it was the practice with only 4 participants. While there is likely to be some self-selecting, this would apply equally to both intervention and control. We have clarified the inclusion criteria by stating that participating practices had to be “able to nominate two GPs to attend the goal-setting training”. We have also added the following clarification to the intervention section “Both intervention and control practices identified two GPs to either attend the training and deliver goal setting consultations or deliver control consultations, although in one intervention practice (Practice

Reviewer	Ref	Section	Original Page	Original Line(s)	Comment	Response
						3) only one GP was able to attend.”
3	98	Methods: Intervention	6	9	Some more minor comments: 1. Page 6, line 14: Authors state 3 questions to consider. I counted 5 (albeit in 3 bullets).	Agreed. We have delete the reference to “three”.
3	99	Methods: Qualitative analysis	8	3	2. Page 8, line 6: “... was used to analyse the focus groups...” – that should say “focus group TRANSCRIPTS”.	Agreed. We have changed this to “A thematic framework-based analysis was used to analyse the focus groups recordings and transcripts”.
3	100	Methods: Qualitative analysis	8	7-12	3. Page 8, line 12: This paragraph would be more clear if it started with a statement of how many PPIs were involved overall and then go into their roles in the project.	Agreed. We have added the statement “Four individuals contributed to patient and public involvement (CG, RH, AM, HS).”
3	101	Results: Outcome measures	10		4. Page 10, lines 4-5: Not clear if this means for all their patients? So the GP that treated it as end of life issues did this for all 3-4 patients they saw as part of the trial?	Yes, this particular GP focused on end of life issues in all of their consultations for the study. We have reworded the first paragraph on page 10 to clarify the characteristics of the control group consultations.
3	102	Discussion	13	14	5. Page 13, line 22: The authors write “were around preparation and agreeing goals”. Is this missing the word “on”?	We have split this sentence into two to make it easier to read. It now reads “Goal setting consultations were more focussed on what matters to the patient than the control consultations. Key challenges in goal setting included preparation and agreeing goals and we explore these further elsewhere [32].”
3	103	Discussion	13	36	6. Page 13, line 49: “patients seeking to agree the desired outcomes of care”. Is this missing the word “to”?	We don’t know where the additional “to” would go, it appears to read fine as is.
3	104	Table 1	n/a	n/a	The manuscript could benefit from some changes to the tables for added clarity. Table 1: Can authors add the number of GPs per practice in total? Also, the row of “characteristics of participating GPs” is quite hard to read. Not clear if that means that for practice one it was 2 male GPs, both were partners and both worked part time? Maybe just another way of noting that would make it easier for the reader to understand quickly.	Unfortunately we don’t have the number of GPs per practice and this data difficult to interpret because of the variety of primary care workforce and work patterns in the UK. We have modified the characteristics of participating GP row to make it easier to read. The ‘Practice recruitment’ data for the three control practices

Reviewer	Ref	Section	Original Page	Original Line(s)	Comment	Response
					And lastly, for the row of “patients assessed for eligibility, n” – the Control numbers are unclear. What is “108(0.6)”? It doesn’t conform to the row title.	appear to have been incorrectly pasted from a previous version of the table, apologies for this. The data, including ‘patients assessed for eligibility’ have now been corrected.
3	105	Table 4	n/a	n/a	Table 4: The title of this table is “Number of goals set and goal attainment score” but that makes it hard to understand the rows titled “Number of goals per patient” – are those numbers by practice the number of people that set those goals? If yes – can this be retitled somehow to add clarity. Otherwise, a reader skimming this table will struggle to understand. Also, can the authors add under “means core of goal attainment per person” what that score is out of (i.e., highest and lowest score)? It would increase readability to have that directly in the table.	We’ve changed the title and edited table 4 to clarify when the numbers refer to patient participants and when they refer to goals. We’ve added in the range to the mean score and given the score for each goal category to aid interpretation.
4	106	Table 6			I am happy with the authors’ responses to my previous comments. I have just a couple of very minor points. Some explanation could be given as to why some of the ICCs are not (cannot?) be reported in Table 6.	We have re-checked the analysis and in fact it was possible to estimate the ICC which happened to be 0.00 in each case. However it was not possible to calculate the confidence interval for these outcomes because the standard error is undefined. We have revised the table to this effect.
4	107	Results	9	31	It is stated in the results that patients spoke more in the goal-setting group, but that other consultation measures “were not statistically significantly higher”. However, looking at Table 3, the estimated between-group difference in WCR has a CI that includes zero.	This is correct that the WCR was not statistically significant, but we would not have expected it to be statistically significant because of the small sample size. Therefore we have added the following caveat to this sentence “but this was not statistically significant”.
5	108	Methods	5	2	As far as I can see the authors have sufficiently addressed the previously comments. I have a few minor follow-on comments: It is stated in the paper that “six month follow-up was long enough for patients and GPs	See response to comment ref 92.

Reviewer	Ref	Section	Original Page	Original Line(s)	Comment	Response
					to work towards goals....” Was this the authors opinion or from somewhere else?	
5	109	Methods: Data & statistical analysis	6	Line 2 of section	Data and Statistical analysis section, page 6 (second line in section): “Data were collected from patients.....”Needs minor rephrasing	Agreed. We’ve amended the text to read “Data collected from patients during a researcher visit at baseline and six months were...”
5	110	Methods: Data & statistical analysis	7	second last para in section	Data and Statistical analysis section, page 7 (: “Key characteristics were compared using a linear mixed model with practice as a random effect”. This appears to be in relation to baseline characteristics (not recommended by consort statement) – please clarify.	No statistical tests were used to compare baseline characteristics. This sentence is a repetition of the following paragraph and refers to statistical testing of the outcomes. We have deleted this sentence.
5	111	Methods: Data & statistical analysis			Data and Statistical analysis section: what statistical package did you use for the analysis?	Stata was used. We’ve added a statement into the methods to read “All statistical analysis were undertaken using Stata version 15.”
5	112	Table 6			Table 6: (1) Need to state that the mean and SD are reported in the table. (2) ICC: “Not estimated” rows, should this be 0.00? Typically the ICC is 0.00 when the between-subject variation is very small compared to within-subject variation.	Agreed. We have added the terms “mean” and “SD” into the table. See response to ref 106 about the ICC.

VERSION 3 - REVIEW

REVIEWER	Agnes Grudniewicz Telfer School of Management, University of Ottawa, Canada
REVIEW RETURNED	27-Mar-2019

GENERAL COMMENTS	The authors did an excellent job addressing all reviewer concerns.
--